# Energy Rank Alignment: Using Preference Optimization to Search Chemical Space at Scale

## Abstract

Searching through chemical space is an exceptionally challenging problem because the number of possible molecules grows combinatorially with the number of atoms. Large, autoregressive models trained on databases of chemical compounds have yielded powerful generators, but we still lack robust strategies for generating molecules with desired properties. This molecular search problem closely resembles the "alignment" problem for large language models, though for many chemical tasks we have a specific and easily evaluable reward function. Here, we introduce an algorithm called energy rank alignment (ERA) that leverages an explicit reward function to produce a gradient-based objective that we use to optimize autoregressive policies. We show theoretically that this algorithm is closely related to proximal policy optimization (PPO) and direct preference optimization (DPO), but has a minimizer that converges to an ideal Gibbs-Boltzmann distribution with the reward playing the role of an energy function. Furthermore, this algorithm is highly scalable, does not require reinforcement learning, and performs well relative to DPO when the number of preference observations per pairing is small. We deploy this approach to align molecular transformers to generate molecules with externally specified properties and find that it does so robustly, searching through diverse parts of chemical space. While our focus here is on chemical search, we also obtain excellent results on an AI supervised task for LLM alignment, showing that the method is scalable and general.

## 1  Introduction

Large language models (LLMs) are trained on large corpora of text to autoregressively generate outputs. These models strongly reflect the distribution of the data on which they are trained [21], and controlling the outputs to reflect externally imposed preferences is an increasingly important challenge for deployment. The aforementioned task, often called "alignment", requires either careful curation of training data or large sets of human preference data—both options are labor-intensive [9]. Reinforcement learning from human feedback (RLHF), a family of algorithms that employs these human preference datasets, has been widely employed to align instruction and chat models [21, 5], but it is both expensive to acquire the training data and difficult to carry out in practice [9]. Recent algorithmic developments, such as direct preference optimization (DPO) [25], simplify the alignment framework by making the reward function implicit, but still require human preference data. While these algorithms succeed in constraining outputs, many "alignment"-like tasks require evaluation that would be difficult for human evaluators.

Generative sampling problems seeking to optimize a reward are common in chemistry, where comparing small molecules using a particular functional assay or computationally accessible property

is often far easier than searching chemical space to identify novel compounds. Recent efforts to build large, domain-specific models for chemistry [10] have shown promising performance on both property prediction and reaction prediction tasks. Nevertheless, just as with LLMs, leveraging these models for molecule optimization requires first guiding "unaligned" models to favor important properties like synthetic accessibility or solubility. Here, we seek to productively search chemical space using transformers by introducing a new preference optimization algorithm, which we call energy rank alignment.

**Our contribution:** We formulate a generic alignment algorithm that we call *Energy Rank Alignment* or ERA that leverages an explicit reward function to guide autoregressive sampling while targeting specific properties or preferences. Unlike reward maximization in RL-based algorithms, the policy that minimizes our objective is designed to sample fluctuations around a maximal reward value to promote sample diversity. Our algorithm enables direct gradient-based optimization of a policy to match the ideal preference distribution and converges asymptotically to an optimal distribution with tuneable entropy and controllable regularization, which we show theoretically. The minimizers of our objective are closely related to the minimizer of PPO and DPO, but we have more direct control over the influence of the regularization relative to fluctuations around the maximum reward. In numerical experiments, we demonstrate that this algorithm successfully aligns a molecule transformer model to identify a highly diverse set of chemicals with properties favored by our choice of reward. Finally, we also show that we obtain competitive performance with ERA on benchmark LLM alignment tasks, but emphasize that the chemical applications are the main focus of this paper.

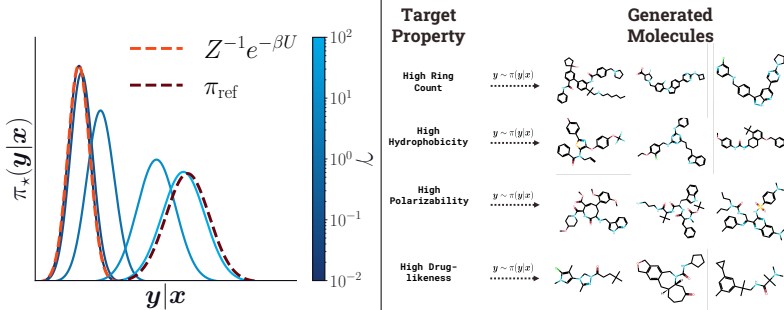

Figure 1: Energy rank alignment (ERA) enables targeting low-energy, high-reward regions with controllable fluctuations. Optimal policy approaches Boltzmann distribution with low regularization ($\gamma \to 0$) and reference policy with high regularization ($\gamma \to \infty$) (left). Aligned models can be used to sample molecules with desired chemical properties (right).

## 1.1 Related Work

Inverse molecular design tasks have a long history [17] and many recent works have sought to apply machine learning to facilitate this difficult search problem [27, 12, 13]. While reinforcement learning has proved a popular strategy for molecular optimization [39, 27], several recent studies have sought to use transformers [34] trained on large databases of molecules represented with the text-based SMILES syntax [10, 30, 35, 4] for such tasks. Schwaller et al. [31] utilized an atom-wise tokenization, which we also employ, to train a transformer for the downstream task of reaction prediction. These "chemical language models" have been studied for applications on downstream tasks, including property prediction [4, 10] and reaction prediction [23, 30].

Building scalable strategies for alignment has attracted enormous attention because of the high cost and complexity of constraining LLM outputs. Much of the current paradigm is built on reinforcement learning from human feedback (RLHF) [21]. Within this framework, human preferences provided in the form of pairwise rankings are first used to train a reward model, and subsequently that reward model is used to optimize a policy using, for example, proximal policy optimization (PPO) [29]. Rafailov et al. [25] demonstrated that the reward model can be treated implicitly using a scheme that maximizes the likelihood of the preferences given an offline dataset. Because this approach does not require training a reward model, it has been named Direct Preference Optimization (DPO). Our work differs from both strategies; first, unlike RLHF, we do not employ reinforcement learning

and instead develop an explicit, gradient-based objective for the optimal policy. Secondly, unlike DPO, we leverage an explicit reward function and add regularization transparently, both of which help to avoid greedy policies [3]. However, like both approaches, we assume that the Bradley-Terry model [7] of preference data is appropriate for the underlying target distribution.

Many recent works have built upon the ideas of RLHF and DPO, including studies on the effect of point-wise sampling of preference distributions [3], investigations into the theoretical basis for contrastive methods for unlearning target datasets [38], and alternatives to the Bradley-Terry pairwise preference model [20, 2]. One recent study explores alignment in the context of inverse molecular design: Park et al. [22] applies DPO to SMILES generators to increase the probability of activity for generated compounds against a drug target. However, they indicate that many preferences in chemistry are expressed as continuous signals, which is not suitable for DPO. Overcoming this limitation while maintaining the advantages of a direct gradient-based policy optimization strategy is a central goal of our current work. Our analysis and methodology directly addresses issues related to point-wise sampling because the explicit reward function eliminates overly greedy assignments of preference probabilities. Indeed, as discussed in Sec. 4, we see that DPO mode collapses where ERA shifts the policy towards the target distribution. While non-transitive preferences may arise in some settings, leading to a breakdown of the Bradley-Terry preference distribution model, by construction our target rewards are determined by quantitative evaluations of properties, and are therefore transitive.

## 2 Energy rank alignment

A policy is a conditional probability distribution $\pi(\cdot|\boldsymbol{x}) : \mathcal{Y} \to \mathbb{R}$; we generate an output $\boldsymbol{y}$ from prompt $\boldsymbol{x}$. The spaces $\mathcal{Y}$ and $\mathcal{X}$ are discrete and finite, corresponding to sequences of tokenized outputs of the model with a maximum length. In alignment tasks, we begin with a pre-trained reference policy $\pi_{\mathrm{ref}}$ and seek to optimize a parametric, trainable policy $\pi_{\boldsymbol{\theta}}$ to adapt the conditional sampling for a particular task or constraint.

Consider a prompt $\boldsymbol{x} \in \mathcal{X}$ and model outputs $\boldsymbol{y}, \boldsymbol{y}' \in \mathcal{Y}$ and a collection of preferences $\mathcal{D} = \{(\boldsymbol{y}_i \succ \boldsymbol{y}'_i; \boldsymbol{x}_i)\}_{i=1}^n$; the notation $\succ$ indicates that $\boldsymbol{y}_i$ is preferred to $\boldsymbol{y}'_i$. The conditional probability that $\boldsymbol{y} \succ \boldsymbol{y}'$ given $\boldsymbol{x}$ can be modeled as a pairwise Boltzmann ranking within the Bradley-Terry model, i.e.,

$$p(\boldsymbol{y} \succ \boldsymbol{y}'|\boldsymbol{x}) = \frac{e^{-\beta U(\boldsymbol{x},\boldsymbol{y})}}{e^{-\beta U(\boldsymbol{x},\boldsymbol{y})} + e^{-\beta U(\boldsymbol{x},\boldsymbol{y}')}} \equiv \sigma\big(\beta U(\boldsymbol{x},\boldsymbol{y}') - \beta U(\boldsymbol{x},\boldsymbol{y})\big). \tag{1}$$

Here $\beta > 0$ is a constant, $\sigma(x) = (1 + e^{-x})^{-1}$ and we refer to $U : \mathcal{X} \times \mathcal{Y} \to \mathbb{R}$ as an energy function to make clear the connection to statistical physics, but it is the negative reward within the RL framework for alignment.

To impose the preferences we minimize the objective

$$J(\pi) = \mathbb{E}_{\boldsymbol{x} \sim \nu} \left[ \int U(\boldsymbol{x},\boldsymbol{y}) \mathrm{d}\pi(\boldsymbol{y}|\boldsymbol{x}) + \beta^{-1} \int (1+\gamma) \log \pi(\boldsymbol{y}|\boldsymbol{x}) - \gamma \log(\pi_{\mathrm{ref}}(\boldsymbol{y}|\boldsymbol{x})) \mathrm{d}\pi(\boldsymbol{y}|\boldsymbol{x}) \right], \tag{2}$$

where $\beta^{-1}$ is a parameter controlling the magnitude of the entropic term, $\gamma$ sets the scale of the Kullback-Leibler regularization compared with the energy term, and $\nu$ is a probability distribution over the prompts $\nu \in \mathcal{P}(\mathcal{X})$. A proximal scheme for gradient descent on this objective corresponds to a gradient flow on $J$ [28, 19]; the functional can be viewed as a free energy, and the corresponding flow is

$$\partial_t \pi_t = \nabla \cdot (\pi_t \nabla \delta_\pi J[\pi_t]), \tag{3}$$

and $\delta_\pi$ denotes the Fréchet derivative with respect to $\pi$. Assuming that $\pi_0$ has full support on $\mathcal{X} \times \mathcal{Y}$, the optimization converges asymptotically to stationary policy which satisfies

$$\nabla \delta_\pi J[\pi_\star] = 0 \iff \pi_\star \propto e^{-\frac{\beta}{1+\gamma} U + \frac{\gamma}{\gamma+1} \log \pi_{\mathrm{ref}}}, \tag{4}$$

and this minimizer is globally optimal. In the context of LLM alignment, a representation of the energy function $U : \mathcal{X} \times \mathcal{Y} \to \mathbb{R}$ is learned as a "reward model", though we also consider tasks in which $U$ is an easily evaluated function of the pair $(\boldsymbol{x}, \boldsymbol{y})$. The optimal distribution $\pi_\star$ is a Gibbs-Boltzmann measure

$$\pi_\star(\boldsymbol{y}|\boldsymbol{x}) = Z^{-1}(\boldsymbol{x}) \exp \left[ -\frac{\beta}{1+\gamma} \big(U(\boldsymbol{x},\boldsymbol{y}) - \beta^{-1}\gamma \log \pi_{\mathrm{ref}}(\boldsymbol{y}|\boldsymbol{x})\big) \right] \tag{5}$$

where $Z(\boldsymbol{x})$ is the $\boldsymbol{x}$-dependent normalization constant. This expression makes clear the effect of $\beta$: when $\beta \to \infty$ (low temperature), the reward dominates and fluctuations around the maximal reward are small, which could lead to "mode-seeking"; when $\beta \to 0$ (high physical temperature) fluctuations around the maximal reward increase and the regularization term favors proximity to $\pi_{\text{ref}}$. Similarly, $\gamma \to 0$ recovers a Gibbs-Boltzmann distribution proportional to $e^{-\beta U}$ at inverse temperature $\beta$, while $\gamma \to \infty$ is dominated by the reference policy.

**Loss functions for $\pi_{\boldsymbol{\theta}}$:** Proximal Policy Optimization (PPO) optimizes an indirect, proximal objective to minimize an objective closely related to (2) (cf. Appendix A). Direct Preference Optimization (DPO) treats the negative reward function $U$ implicitly and directly maximizes the likelihood of $p(\boldsymbol{y} \succ \boldsymbol{y}'|\boldsymbol{x})$. Our objectives differ from both approaches: like DPO, we directly optimize the policy using an explicit, gradient-based objective, but, in contrast, we use a reward function directly in our objective. The losses we build are thus amenable to both offline (samples from $\pi_{\text{ref}}$) and online (samples from $\pi_{\boldsymbol{\theta}}$) policy alignment, as explained below. Choosing to optimize the objective online has been shown to have important consequences on performance [32], though we focus here on the setting where samples are drawn offline.

We directly optimize the Kullback-Leibler divergence between the entropy-regularized preference distribution $p_\gamma(\boldsymbol{y} \succ \boldsymbol{y}'|\boldsymbol{x})$ and the corresponding parametric preference distribution $p_{\boldsymbol{\theta}}(\boldsymbol{y} \succ \boldsymbol{y}'|\boldsymbol{x})$. Explicitly, using the fact that conditional preference distribution is normalized, we obtain

$$
\begin{aligned}
D_{\text{KL}}^{(\boldsymbol{y},\boldsymbol{y}')}(p_\gamma|p_{\boldsymbol{\theta}}) &= p_\gamma(\boldsymbol{y} \succ \boldsymbol{y}'|\boldsymbol{x}) \log \frac{p_\gamma(\boldsymbol{y} \succ \boldsymbol{y}'|\boldsymbol{x})}{p_{\boldsymbol{\theta}}(\boldsymbol{y} \succ \boldsymbol{y}'|\boldsymbol{x})} + p_\gamma(\boldsymbol{y}' \succ \boldsymbol{y}|\boldsymbol{x}) \log \frac{p_\gamma(\boldsymbol{y}' \succ \boldsymbol{y}|\boldsymbol{x})}{p_{\boldsymbol{\theta}}(\boldsymbol{y}' \succ \boldsymbol{y}|\boldsymbol{x})}, \\
&= p_\gamma(\boldsymbol{y} \succ \boldsymbol{y}'|\boldsymbol{x}) \log \frac{p_\gamma(\boldsymbol{y} \succ \boldsymbol{y}'|\boldsymbol{x})}{p_{\boldsymbol{\theta}}(\boldsymbol{y} \succ \boldsymbol{y}'|\boldsymbol{x})} + \big(1 - p_\gamma(\boldsymbol{y} \succ \boldsymbol{y}'|\boldsymbol{x})\big) \log \frac{1 - p_\gamma(\boldsymbol{y} \succ \boldsymbol{y}'|\boldsymbol{x})}{1 - p_{\boldsymbol{\theta}}(\boldsymbol{y} \succ \boldsymbol{y}'|\boldsymbol{x})},
\end{aligned}
\tag{6}
$$

where

$$
p_\gamma := \sigma\left( \frac{\beta}{1+\gamma} \left[ (U(\boldsymbol{x},\boldsymbol{y}') - U(\boldsymbol{x},\boldsymbol{y})) + \beta^{-1}\gamma \log \frac{\pi_{\text{ref}}(\boldsymbol{y}|\boldsymbol{x})}{\pi_{\text{ref}}(\boldsymbol{y}'|\boldsymbol{x})} \right] \right).
\tag{7}
$$

This quantity is a well-defined KL divergence and is hence non-negative; the quantity vanishes when $p_\gamma = p_{\boldsymbol{\theta}}$ on the observations $\boldsymbol{y}, \boldsymbol{y}'$. Furthermore, with access to an explicit reward model, all terms in (6) can be computed directly and

$$
p_{\boldsymbol{\theta}}(\boldsymbol{y} \succ \boldsymbol{y}'|\boldsymbol{x}') = \frac{\pi_{\boldsymbol{\theta}}(\boldsymbol{y}|\boldsymbol{x})}{\pi_{\boldsymbol{\theta}}(\boldsymbol{y}|\boldsymbol{x}) + \pi_{\boldsymbol{\theta}}(\boldsymbol{y}'|\boldsymbol{x})} = \sigma\left( \log \frac{\pi_{\boldsymbol{\theta}}(\boldsymbol{y}|\boldsymbol{x})}{\pi_{\boldsymbol{\theta}}(\boldsymbol{y}'|\boldsymbol{x})} \right).
\tag{8}
$$

To obtain a minimizer of the regularized objective defined in (2) we optimize

$$
\mathcal{L}^{\text{ERA}}(\pi_{\boldsymbol{\theta}}) = \mathbb{E}_{x \sim \mathcal{D}} \mathbb{E}_{\boldsymbol{y},\boldsymbol{y}' \sim \pi_{\text{ref}}(\cdot|\boldsymbol{x})} D_{\text{KL}}^{(\boldsymbol{y},\boldsymbol{y}')}(p_\gamma|p_{\boldsymbol{\theta}});
\tag{9}
$$

If the current policy overlaps with the target preference distribution, it may be useful to sample directly from the partially aligned policy, i.e., to use the "on-policy" formulation,

$$
\mathcal{L}_{\text{on}}^{\text{ERA}}(\pi_{\boldsymbol{\theta}}) = \mathbb{E}_{\boldsymbol{x} \sim \mathcal{D}} \mathbb{E}_{\boldsymbol{y},\boldsymbol{y}' \sim \pi_{\boldsymbol{\theta}}(\boldsymbol{y}|\boldsymbol{x})} D_{\text{KL}}^{(\boldsymbol{y},\boldsymbol{y}')}(p_\gamma|p_{\boldsymbol{\theta}})
\tag{10}
$$

instead of (9). One issue that arises with this scheme is that differentiation with respect to the parameters of the policy $\boldsymbol{\theta}$ because $\boldsymbol{y}$ and $\boldsymbol{y}'$ are decoded into discrete tokens, an operation that is not differentiable. To remedy this, we importance sample with a reference policy

$$
\mathcal{L}_{\text{on}}^{\text{ERA}}(\pi_{\boldsymbol{\theta}}) = \mathbb{E}_{\boldsymbol{x} \sim \mathcal{D}} \mathbb{E}_{\boldsymbol{y},\boldsymbol{y}' \sim \pi_{\text{ref}}(\boldsymbol{y}|\boldsymbol{x})} \frac{\pi_{\boldsymbol{\theta}}(\boldsymbol{y}|\boldsymbol{x})\pi_{\boldsymbol{\theta}}(\boldsymbol{y}'|\boldsymbol{x})}{\pi_{\text{ref}}(\boldsymbol{y}|\boldsymbol{x})\pi_{\text{ref}}(\boldsymbol{y}'|\boldsymbol{x})} D_{\text{KL}}^{(\boldsymbol{y},\boldsymbol{y}')}(p_\gamma|p_{\boldsymbol{\theta}}).
\tag{11}
$$

This reweighting is straightforward and the importance weights should generally be appreciable, especially early in training when $\pi_{\boldsymbol{\theta}}$ has not drifted far from $\pi_{\text{ref}}$. It is, of course, also natural to iteratively update $\pi_{\boldsymbol{\theta}}$ using a previous iterate as the reference policy. In this work, we only use (9) as an objective and leave the on-policy objectives to future work.

## 3 Theoretical Analysis

To understand the ERA loss function and its connection to the entropy regularized objective (2), we first establish that the minimizers of (6) are of the form (5). We first define the notion of equivalence precisely.

**Definition 3.1** *The conditional probability measures $\pi(\cdot|\boldsymbol{x})$ and $\pi'(\cdot|\boldsymbol{x})$ are conditionally equivalent if $\forall \boldsymbol{x} \in \mathcal{X}$, $\pi$ and $\pi'$ are such that $\sup_{\boldsymbol{y} \in \mathcal{Y}} |\pi(\boldsymbol{y}|\boldsymbol{x}) - \pi'(\boldsymbol{y}|\boldsymbol{x})| = 0$.*

We remark that this strong form of equivalence is appropriate on the finite, discrete spaces $\mathcal{X}$ and $\mathcal{Y}$ we consider here.

**Lemma 3.1** *If $\pi$ is conditionally equivalent to $\pi'$, then $\pi'_g(\cdot|\boldsymbol{x}) \propto \pi'(\cdot|\boldsymbol{x})e^{g(\boldsymbol{x})}$ is conditionally equivalent to $\pi$ for all functions $g : \mathcal{X} \to \mathbb{R}$ such that $\sup_{\boldsymbol{x} \in \mathcal{X}} |e^{g(\boldsymbol{x})}| < +\infty$.*

We prove Lemma 3.1 in Appendix A and use this simple lemma to prove the following result.

**Proposition 3.2** *Suppose $\pi(\cdot|\boldsymbol{x}) \in \mathcal{P}(\mathcal{Y})$ and that $\mathrm{supp}(\pi) = \mathrm{supp}(\pi_{\mathrm{ref}})$. Let $\beta > 0$, $\gamma \geq 0$ and that the reward model is such that $\sup_{\boldsymbol{x},\boldsymbol{y} \in \mathcal{X} \times \mathcal{Y}} |e^{-U(\boldsymbol{x},\boldsymbol{y})}| < +\infty$. Then, the minimizer of $\mathcal{L}^{\mathrm{ERA}}$ is conditionally equivalent to $\pi_{\star}$.*

First, we verify that any probability measure $\pi_g(\boldsymbol{y}|\boldsymbol{x}) \propto \exp(-\frac{\beta}{1+\gamma}\big(U(\boldsymbol{x},\boldsymbol{y}) - \beta^{-1}\gamma \log \pi_{\mathrm{ref}}(\boldsymbol{y}|\boldsymbol{x})\big) + g(\boldsymbol{x}))$ minimizes the objective. Because $\mathcal{L}^{\mathrm{ERA}}$ is non-negative, it suffices to show that for all pairs $\boldsymbol{y}, \boldsymbol{y}'$, $D_{\mathrm{KL}}^{(\boldsymbol{y},\boldsymbol{y}')}(p_\gamma|p_{\boldsymbol{\theta}}) \equiv 0$. This follows immediately from the cancellation in the preference probability $p_\gamma$ of $e^{g(\boldsymbol{x})}$ after factorization in (5). Now, suppose that $\pi(\boldsymbol{y}|\boldsymbol{x}) \neq \exp\left(-\frac{\beta}{1+\gamma}\big(U(\boldsymbol{x},\boldsymbol{y}) - \beta^{-1}\gamma \log \pi_{\mathrm{ref}}(\boldsymbol{y}|\boldsymbol{x})\big)\right)$ where we have taken $g(\boldsymbol{x}) = 0$ without loss of generality and $\pi := \pi_g$. Assume that for all pairs $\boldsymbol{y}, \boldsymbol{y}'$, the divergence $D_{\mathrm{KL}}^{(\boldsymbol{y},\boldsymbol{y}')}(p_\gamma|p_{\boldsymbol{\theta}}) \equiv 0$ which is required of a minimizer. Equivalently, it must be the case that for all $\boldsymbol{y}, \boldsymbol{y}'$,

$$\frac{\pi(\boldsymbol{y}|\boldsymbol{x})}{\pi(\boldsymbol{y}|\boldsymbol{x}) + \pi(\boldsymbol{y}'|\boldsymbol{x})} = \frac{\pi_{\star}(\boldsymbol{y}|\boldsymbol{x})}{\pi_{\star}(\boldsymbol{y}|\boldsymbol{x}) + \pi_{\star}(\boldsymbol{y}'|\boldsymbol{x})} \implies \frac{\pi(\boldsymbol{y}'|\boldsymbol{x})}{\pi(\boldsymbol{y}|\boldsymbol{x})} = \frac{\pi_{\star}(\boldsymbol{y}'|\boldsymbol{x})}{\pi_{\star}(\boldsymbol{y}|\boldsymbol{x})}, \qquad (12)$$

from which we see that

$$\pi(\boldsymbol{y}|\boldsymbol{x}) = \frac{\pi(\boldsymbol{y}'|\boldsymbol{x})}{e^{-\frac{\beta}{1+\gamma}(U(\boldsymbol{x},\boldsymbol{y}') - \beta^{-1}\gamma \log \pi_{\mathrm{ref}}(\boldsymbol{y}'|\boldsymbol{x}))}} e^{-\frac{\beta}{1+\gamma}\big(U(\boldsymbol{x},\boldsymbol{y}) - \beta^{-1}\gamma \log \pi_{\mathrm{ref}}(\boldsymbol{y}|\boldsymbol{x})\big)}. \qquad (13)$$

By construction, $\pi(\boldsymbol{y}|\boldsymbol{x})$ does not depend on $\boldsymbol{y}'$ so the prefactor must be purely a function of $\boldsymbol{x}$, which completes the proof, using Lemma 3.1.

**Gradients of $\mathcal{L}^{\mathrm{ERA}}$.** One advantage of the ERA framework is that the objective is amenable to direct, gradient-based optimization. We remark that establishing global convergence for the optimization of $\boldsymbol{\theta}$ using (9) requires establishing convexity with respect to the parameters, which is not obviously the case for our objective, nor those used in PPO and DPO. However, one can still glean some insight into the optimization by examining the gradients on a samplewise basis. Using the compact notation $p_{\boldsymbol{\theta}}(\boldsymbol{y} \succ \boldsymbol{y}'|\boldsymbol{x}) \equiv \sigma_{\boldsymbol{\theta}}$ and $p_\gamma(\boldsymbol{y} \succ \boldsymbol{y}'|\boldsymbol{x}) \equiv \sigma_{\star}$,

$$\nabla_{\boldsymbol{\theta}} \mathcal{L}^{\mathrm{ERA}} = \mathbb{E}_{\boldsymbol{x} \sim \mathcal{D}} \mathbb{E}_{\boldsymbol{y},\boldsymbol{y}' \sim \pi_{\mathrm{ref}}} \left(\frac{1 - \sigma_{\star}}{1 - \sigma_{\boldsymbol{\theta}}} - \frac{\sigma_{\star}}{\sigma_{\boldsymbol{\theta}}}\right) \nabla_{\boldsymbol{\theta}} \sigma_{\boldsymbol{\theta}}. \qquad (14)$$

The gradient is straightforward to interpret on a particular pair $\boldsymbol{y}, \boldsymbol{y}'$: if $p_{\boldsymbol{\theta}}(\boldsymbol{y} \succ \boldsymbol{y}'|\boldsymbol{x})$ is larger than $p_\gamma(\boldsymbol{y} \succ \boldsymbol{y}'|\boldsymbol{x})$ then the preference gradient is positive and gradient descent lowers the probability that $\boldsymbol{y} \succ \boldsymbol{y}'$. The opposite occurs whenever $p_{\boldsymbol{\theta}}(\boldsymbol{y} \succ \boldsymbol{y}'|\boldsymbol{x})$ is smaller than $p_\gamma(\boldsymbol{y} \succ \boldsymbol{y}'|\boldsymbol{x})$. The magnitude of the gradient is scaled by the degree of misspecification of the preference probability.

This calculation highlights one key difference between the approach we use and DPO. When the data only contains one observation of $\boldsymbol{y} \succ \boldsymbol{y}'$ for a given $\boldsymbol{x}$, the DPO objective's implicit reward model assigns zero probability to $\boldsymbol{y}' \succ \boldsymbol{y}$. This pushes the policy towards extremal values, which can lead to undesired behavior, as discussed in Azar et al. [3]. In our formulation, this behavior occurs only when the reward model assigns an energy of $\pm\infty$, which is prohibited by construction in most tasks. We further discuss differences between ERA and DPO in Appendix A.2.

# 4 Experiments

We test ERA on both chemical and language tasks to shed light on the following questions: 1) Can we use ERA to robustly fine-tune our model to generate samples according to a desired distribution?

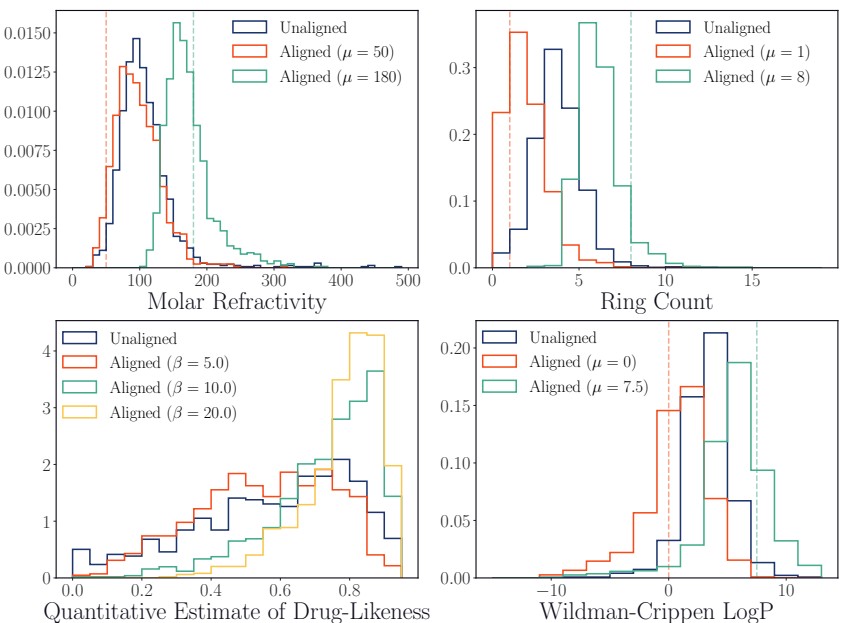

Figure 2: Unprompted molecular generator alignment. Distributions of different chemical properties for molecules sampled from aligned and unaligned policies. The center of the harmonic potential, $\mu$, is varied for MR ($\beta = 1.0$), Ring Count ($\beta = 1.0$), and LogP ($\beta = 10.0$), while $\beta$ is varied for QED. All experiments were run with no regularization to the reference policy ($\gamma = 0$).

2) What is the effect of changing the inverse-temperature $\beta$ during ERA? 3) Do we maintain sample diversity (and validity) without regularizing to remain close to a reference policy, and what is the effect of increased regularization? 4) Can we simultaneously target multiple properties with high fidelity, and how can we trade off between desired properties? 5) Can we carry out ERA on higher capacity models with "weak" signals from smaller models?

### 4.1 Generating molecules with desired properties

We use a decoder-only representation for the molecular generator [4], where the generator has 2 layers, an embedding dimension of 512, a vocabulary of 324 tokens, and totals 3.5M parameters. Starting from a random initialization, we carry out pretraining on a dataset of 2.4M small molecules from the ChEMBL database [37] for 180 epochs. This version of the model is not conditioned on a prompt and generates a small molecule given just a start-of-sequence token. We use this pretrained model as our reference policy for all unprompted molecular alignment tasks (Sec. 4.1.1). In Sec. 4.1.2, we generate molecules conditioned on a prompt using a generator that was trained to carry out sampling with a prompt molecule.

Central to ERA is, of course, access to a computable energy function. As a proof-of-concept, here we consider 5 different properties for which the corresponding energy function is easily evaluable: Quantitative Estimate of Drug-Likeness (QED) [6], Wildman-Crippen LogP (LogP) [36], Ring Count, Molar Refractivity (MR) [36], and Tanimoto Similarity [26]. Briefly, LogP is a measure of the hydrophobicity of a molecule, MR is a measure of the polarizability of the molecule, and Tanimoto similarity is a measure of the similarity between two molecules (see Appendix C.2).

#### 4.1.1 Unprompted molecular alignment

First, we independently target four different properties using ERA with an unprompted molecular generator (Fig. 2). Using the reference policy, we generate a dataset $\mathcal{D} = \{\boldsymbol{y}_1^{(i)}, \boldsymbol{y}_2^{(i)}, U(\boldsymbol{y}_1^{(i)}), U(\boldsymbol{y}_2^{(i)})\}_{i=1}^N$ and carry out energy rank alignment on $\pi_{\boldsymbol{\theta}}$, where $\pi_{\boldsymbol{\theta}}$ is initialized using the weights of $\pi_{\text{ref}}$. Here, $\boldsymbol{y}_1, \boldsymbol{y}_2 \sim \pi_{\text{ref}}$ and $\boldsymbol{y}$ and $U(\boldsymbol{y})$ denote the generated molecule and its corresponding energy, respectively. For MR, Ring Count, and LogP, we define the energy $U$ to be

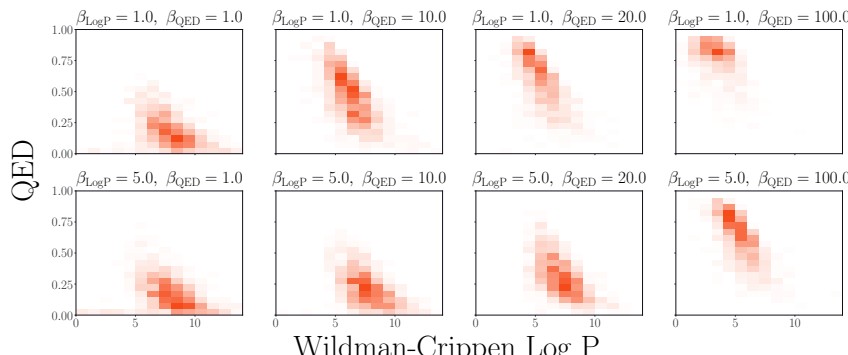

Figure 3: Unprompted multi-property molecular generator alignment. 2D histograms of LogP versus QED for different combinations of property-specific $\beta$ illustrating a clear trade-off when performing multi-property alignment. Relative increases in $\beta$ for a given property target higher values for that property. All experiments were run with no regularization to the reference policy ($\gamma = 0$).

a harmonic potential centered at a target value. For QED, we define the energy to be the negative logarithm of QED and vary $\beta$ to assess its impact on alignment (see Table 1, 2). In Fig. 2, we see that we successfully shift the distribution to target means that are both greater and lower than the average value of MR, Ring Count, and LogP under the reference policy. Furthermore, in the alignment of QED, we observe the effect of changing $\beta$ on the learned policy; with increased $\beta$, the learned policy concentrates around low-energy samples (i.e. near QED = 1), and with lower $\beta$, the learned policy samples a greater range of QED values, as expected. We note that for each of these four experiments, we did not regularize towards the reference policy (i.e. $\gamma = 0$). Even so, we were able to maintain both sample diversity and maintain appreciable sample validity (see Fig. 7 and Table 3).

Many molecular design tasks require balancing multiple properties, and designing an objective for multi-property alignment is straightforward within the ERA framework. To demonstrate this, we generate molecules with both high QED and LogP using ERA with an energy function weighted by property-specific $\beta$: $U = \beta_{\text{QED}} U_{\text{QED}} + \beta_{\text{LogP}} U_{\text{LogP}}$ (see Table 1, 4 for details on energy function). We carry out ERA with different pairs of $(\beta_{\text{QED}}, \beta_{\text{LogP}})$ using the same procedure as above, and from Fig. 3, we see that we target multiple properties with varying fidelity by simply modulating the value of property-specific $\beta$. Ultimately, increasing the $\beta$ for an individual property enables us to favor higher values of that property in multi-property alignment setting. In this case, we also do not regularize with the KL-divergence to the reference policy and again maintain sample diversity and validity (see Fig. 8 and Table 4)

### 4.1.2 Prompted molecular alignment

Inspired by the task of lead optimization in drug discovery efforts [16], we ask whether we can use ERA to train a molecular generator that can sample a molecule that is both similar to the prompt molecule *and* also exhibits some desired property.

First, we fine-tune the pretrained molecular generator to enable prompted molecular generation (see Appendix C.3.2) and use this fine-tuned model as our reference policy for all prompted molecular alignment tasks. This reference policy disproportionately samples molecules that are identical (i.e. a Tanimoto similarity of 1.0) to the prompt molecule (see Fig. 4), so we carry out multi-property alignment on this reference policy to generate molecules that are similar—but not identical—to the prompt molecule and also have a high drug-likeness as measured by QED. Using ERA, we optimize the reference policy with a generated dataset $\mathcal{D} = \{(\boldsymbol{y}_1^{(i)}, \boldsymbol{x}^{(i)}), (\boldsymbol{y}_2^{(i)}, \boldsymbol{x}^{(i)}), U(\boldsymbol{y}_1^{(i)}, \boldsymbol{x}^{(i)}), U(\boldsymbol{y}_2^{(i)}, \boldsymbol{x}^{(i)})\}_{i=1}^N$, where we sample four molecules for each prompt molecule from the reference policy and consider all possible preference pairs for a total of six preference pairs per prompt molecule (see Appendix C.2 for full details on energy used).

We observe that the per-prompt average QED under the optimized policy for a given prompt is higher than the corresponding average under the reference policy (Fig. 4). Furthermore, we see that we are able to sample a diverse set of molecules that are chemically similar to the prompt molecule, and

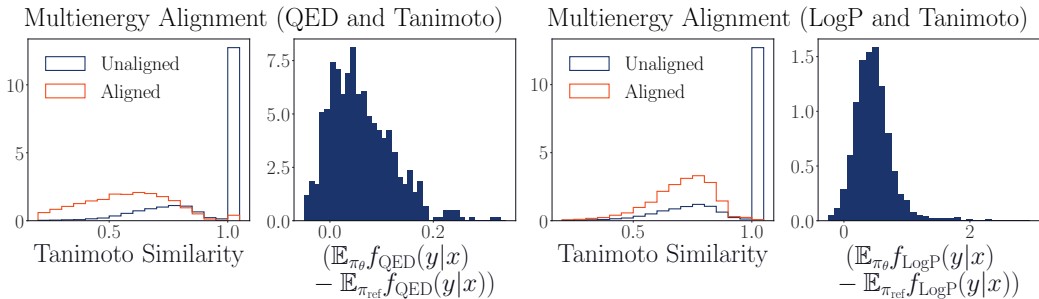

Figure 4: Prompted multi-property molecular generator alignment. From left to right: Tanimoto similarities computed between the prompt and sampled molecules for both aligned and unaligned policies (QED and Tanimoto alignment), per-prompt difference in the average QED under aligned and unaligned policies (QED and Tanimoto alignment), Tanimoto similarities computed between the prompt and sampled molecules for both aligned and unaligned policies (LogP and Tanimoto alignment), and per-prompt difference in the average LogP under aligned and unaligned policies (LogP and Tanimoto alignment). With alignment, we target higher QED and LogP values, while still sampling molecules chemically similar—but not identical—to prompt molecule.

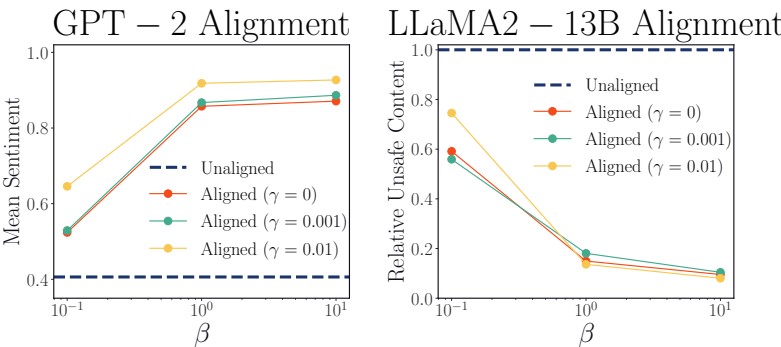

Figure 5: AI-guided alignment of LLMs. Average sentiment of responses from aligned GPT-2 model across all prompts. (left). Proportion of unsafe content relative to unaligned model of responses aligned LLaMA2-13B model across all prompts (right). 5.4% of all responses from unaligned model were classified as unsafe. Error bars too small to be shown.

also chemically valid (see Figure 9, Table 5). We repeat the experiment with a related objective of generating molecules similar to the prompt molecule with a high LogP instead and again observe that we increase the per-prompt average LogP under the optimized policy relative to the reference policy without degrading sample diversity and validity. For both of these experiments, we required regularization to the reference policy. With no regularization, the aligned generator would almost exclusively sample sequences that were chemically invalid ($< 25\%$ chemical validity). Finally, we note that the increases in QED and LogP in Fig. 4 are smaller relative to the increases in Fig. 2 because the samples are now conditioned to remain proximal to the prompt molecule, which restricts the chemical space that can be explored.

## 4.2 AI-guided alignment of large language models

We test the generality of ERA by applying it to align large language models (LLMs). Similar to the experiments in [25], we first carry out ERA on a GPT-2 model [24] fine-tuned on movies reviews from IMDb [18]. We use a pretrained sentiment classifier [14] to evaluate the energies—where lower energies correspond to more positive sentiments—of sampled responses from the reference policy and carry out ERA using the same approach as in Section 4.1.2 (see Appendix D.1). We vary the regularization strength $\gamma$ and inverse-temperature $\beta$ on the average sentiment and observe that across all regularization strengths, with increasing $\beta$, the average sentiment becomes more positive. Increasing regularization also elicits more positive sentiments. Qualitatively, with lower

regularization, we observe that text quality degrades and becomes less coherent, likely resulting in lower average sentiment predictions by the sentiment model. Regularization here is important to ensure high quality text samples.

We next leverage a "weak" AI supervisor to carry out LLM alignment, a task sometimes called "superalignment" [8]. In the present context, we order "weak" vs. "strong" models based on their parameter count (within the same family) and empirical performance; i.e., LLaMA2-7B is weaker than LLaMA2-13B. Here, the weak model does not necessarily contain the complexity of the stronger model but can *weakly* discern between different outputs of a stronger model. Given a sample $\boldsymbol{y}_i \sim \pi_{\text{strong}}(\boldsymbol{y}|\boldsymbol{x})$, we define the energy using the weak model $U(\boldsymbol{y}_i|\boldsymbol{x}) = -\log \pi_{\text{weak}}(\boldsymbol{y}_i|\boldsymbol{x})$.

We test *weak-to-strong alignment* using a previously aligned LLaMA2-7B-Chat (meta-llama/Llama-2-7b-chat) to optimize an unaligned LLaMA2-13B (meta-llama/Llama-2-13b) model [33]. Using prompts from the Anthropic Helpful and Harmless dialogue dataset [5], we first carry out a short supervised fine-tuning step of LLaMA2-13B to ensure it can output text in a chat-like format (see Appendix D.2). Using this reference policy, we generate a dataset with energies computed from the smaller LLaMA2-7B-Chat model and carry out ERA as above, again across varying $\gamma$ and $\beta$. We evaluate the "safety" of generated samples using Meta LLama Guard 2 (meta-llama/Meta-Llama-Guard-2-8B) [15]. We observe that as we increase $\beta$, the proportion of unsafe content relative to the unaligned, reference model decreases, with over a 90% drop between the unaligned model and the models aligned with the highest $\beta$ across all $\gamma$. For these experiments, we observe that varying regularization strengths has a minimal effect and that we are in fact able to generate coherent sentences with no regularization, with strong regularization hurting performance for $\beta = 0.1$. Finally, we compare ERA and DPO in Appendix D.2 and observe that with our implementation of DPO, we are able to generate lower energy samples, but that it is prone to mode collapse. We caution that our implementation of DPO is likely not optimal and that we did not exhaustively tune the hyperparameters of DPO due to resource constraints.

## 5 Conclusions and Limitations

This paper introduces energy rank alignment, a simple and effective algorithm for policy optimization with an explicit reward model. We find that ERA is stable without extensive hyperparameter tuning, and sufficiently general to successfully align both application-specific transformers for chemical search problems as well as generative pre-trained transformers for language. The algorithm exhibits strong performance with a variety of reward models, even ones with relatively weak signal, such as the AI feedback of LLaMA2-7B-Chat. Interestingly, with this approach we are able to reduce unsafe content by more than 90% with no human preference data.

We analyze the minimizers of the ERA objective and find that they differ from the minimizers of popular policy alignment algorithms DPO and PPO in an important way: unlike PPO, the strength of regularization to the reference policy that we add is controlled by a parameter $\gamma$, while the entropy of the target distribution is independently tuned by a distinct parameter $\beta$. This means that we can avoid greedy policies by keeping $\beta$ small—amplifying fluctuations around the optimum of the reward model $-U$—while reducing the influence of the reference policy by taking $\gamma$ small. Our objective leads to easily interpretable sample-wise gradients which highlight the importance of a reward model relative to DPO in the sampled objective. Similar observations about the inadequacy of the DPO objective for finite preference observations were also made theoretically in Azar et al. [3].

**Limitations:** First, our approach requires a reward model, which can be difficult to train or design, especially for complex tasks. While we observed that ERA makes an appreciable impact even with weak supervision from an AI chat model, this sort of proxy may not be available for more complex tasks. For example, optimizing small molecules for high binding affinity to a target protein would require expensive and noisy evaluations of a reward model, which likely limits the scope of molecular design to problems where the reward can be computed somewhat efficiently. A second limitation of our present work is that we do not train the molecular transformer to favor synthetic accessibility nor do we explicitly seek to obtain molecules that are easily synthesized experimentally. There are models that seek to evaluate synthesizability computationally that could be used in our rewards, which we plan to explore in future work [11]. A final limitation of our current work is the moderate scale of our numerical experiments due to our limited compute resources, including the inadequate hyperparameter tuning for the DPO baseline for Fig. 5.

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

## A Detailed Theoretical Analysis

**Set-up, notation, and assumptions** Let $\mathcal{X}$ and $\mathcal{Y}$ be discrete spaces; each element of one of these spaces is a finite-length sequence of tokens within a fixed dictionary on which an autoregressive generative model is trained. The resulting models yield "policies", which are conditional probability distributions $\pi(\cdot|\boldsymbol{x}) \in \mathcal{P}(\mathcal{Y})$ for each $\boldsymbol{x} \in \mathcal{X}$. Throughout, we assume that our policies have full support on $\mathcal{Y}$ for each $\boldsymbol{x}$, meaning that $\inf_{\boldsymbol{y},\boldsymbol{x} \in \mathcal{Y} \times \mathcal{X}} \pi(\boldsymbol{y}|\boldsymbol{x}) > 0$. Because the spaces are discrete, we make no strong restrictions on the regularity or coerciveness of the reward model $-U : \mathcal{X} \times \mathcal{Y} \to \mathbb{R}$. The only requirement to ensure the existence of an optimal probability distribution is that $\sup_{\boldsymbol{x},\boldsymbol{y} \times \mathcal{X} \times \mathcal{Y}} |e^{-U(\boldsymbol{x},\boldsymbol{y})}| < +\infty$, which maintains full support of the distribution. Though it plays little role in theoretical analysis, we also denote by $\nu \in \mathcal{P}(\mathcal{X})$ the probability distribution over the prompts $\boldsymbol{x}$.

**Goals of the analysis presented here** The main purpose of this section is to establish that globally minimizing the loss (9) yields a global minimizer of the regularized policy objective (2). A secondary goal is to clearly articulate the theoretical advantages of ERA compared with PPO and DPO.

To understand the ERA loss function and its connection to the entropy regularized objective (2), we first establish that the minimizer of (6) are of the form (5). We first define the notion of equivalence precisely.

**Definition A.1** *The conditional probability measures $\pi(\cdot|\boldsymbol{x})$ and $\pi'(\cdot|\boldsymbol{x})$ in $\mathcal{P}(\mathcal{Y})$ are conditionally equivalent if $\forall \boldsymbol{x} \in \mathcal{X}$, $\pi$ and $\pi'$ are such that $\sup_{\boldsymbol{y} \in \mathcal{Y}} |\pi(\boldsymbol{y}|\boldsymbol{x}) - \pi'(\boldsymbol{y}|\boldsymbol{x})| = 0$.*

This is a strong form of equivalence for probability measures, but it is appropriate on the discrete spaces $\mathcal{X}$ and $\mathcal{Y}$ we consider here. For more general continuous spaces, one could relax this condition to weak equivalence of the conditional measures. We use this notion to emphasize that a shift of the distribution of the "prompts" $\boldsymbol{x} \in \mathcal{X}$, which we denote $\nu \in \mathcal{P}(\mathcal{X})$, does not impact conditional equivalence and hence establishes an equivalence class of conditional probability measures that minimize (2).

**Lemma A.1** *If $\pi$ is conditionally equivalent to $\pi'$, then $\pi'_g(\cdot|\boldsymbol{x}) \propto \pi'(\cdot|\boldsymbol{x})e^{g(\boldsymbol{x})}$ is conditionally equivalent to $\pi$ for all functions $g : \mathcal{X} \to \mathbb{R}$ such that $\sup_{\boldsymbol{x} \in \mathcal{X}} |e^{g(\boldsymbol{x})}| < +\infty$.*

Assume that $\pi'$ is a normalized probability distribution. This requires that,

$$Z'(\boldsymbol{x}) = \sum_{\boldsymbol{y} \in \mathcal{Y}} \pi'(\boldsymbol{y}|\boldsymbol{x}) = 1. \tag{15}$$

If $g$ is such that

$$Z'_g(\boldsymbol{x}) = \sum_{\boldsymbol{y} \in \mathcal{Y}} \pi'(\boldsymbol{y}|\boldsymbol{x})e^{g(\boldsymbol{x})} \neq 1, \tag{16}$$

then the normalized policy $\pi'_g$ is clearly defined by

$$\frac{1}{Z'_g(\boldsymbol{x})}\pi'(\boldsymbol{y}|\boldsymbol{x})e^{g(\boldsymbol{x})} \equiv \pi'(\boldsymbol{y}|\boldsymbol{x}), \tag{17}$$

because $Z'_g(\boldsymbol{x}) = e^{g(\boldsymbol{x})}$. By the assumption that $\sup_{\boldsymbol{x} \in \mathcal{X}} |e^{g(\boldsymbol{x})}| < +\infty$, all terms in these calculations remain finite.

Using Lemma A.1 it is straightforward to prove the result in the main text Proposition 3.2. For completeness, we re-state that result here and refer the reader to the main text for the complete argument.

**Proposition A.2** *Suppose $\pi(\cdot|\boldsymbol{x}) \in \mathcal{P}(\mathcal{Y})$ and that $\mathrm{supp}(\pi) = \mathrm{supp}(\pi_{\mathrm{ref}})$. Let $\beta > 0$, $\gamma \geq 0$ and that the reward model is such that $\sup_{\boldsymbol{x},\boldsymbol{y} \in \mathcal{X} \times \mathcal{Y}} |e^{-U(\boldsymbol{x},\boldsymbol{y})}| < +\infty$. Then, the minimizer of $\mathcal{L}^{\mathrm{ERA}}$ is conditionally equivalent to $\pi_{\star}$.*

This proposition establishes that a policy minimizing the objective

$$
\begin{aligned}
\mathcal{L}^{\mathrm{ERA}}(\pi_{\boldsymbol{\theta}}) &= \mathbb{E}_{x\sim\mathcal{D}}\mathbb{E}_{\boldsymbol{y},\boldsymbol{y}'\sim\pi_{\mathrm{ref}}(\cdot|\boldsymbol{x})} D_{\mathrm{KL}}^{(\boldsymbol{y},\boldsymbol{y}')}(p_{\beta}|p_{\boldsymbol{\theta}}); \\
p_{\boldsymbol{\theta}} &:= \sigma\left(\log\frac{\pi_{\boldsymbol{\theta}}(\boldsymbol{y}|\boldsymbol{x})}{\pi_{\boldsymbol{\theta}}(\boldsymbol{y}'|\boldsymbol{x})}\right) \\
p_{\gamma} &:= \sigma\left(\frac{\beta}{1+\gamma}\left[(U(\boldsymbol{x},\boldsymbol{y}')-U(\boldsymbol{x},\boldsymbol{y}))+\beta^{-1}\gamma\log\frac{\pi_{\mathrm{ref}}(\boldsymbol{y}|\boldsymbol{x})}{\pi_{\mathrm{ref}}(\boldsymbol{y}'|\boldsymbol{x})}\right]\right),
\end{aligned}
\tag{18}
$$

has the form

$$
\pi_{\star}(\boldsymbol{y}|\boldsymbol{x}) = Z^{-1}(\boldsymbol{x})\exp\left[-\frac{\beta}{1+\gamma}\big(U(\boldsymbol{x},\boldsymbol{y})-\beta^{-1}\gamma\log\pi_{\mathrm{ref}}(\boldsymbol{y}|\boldsymbol{x})\big)\right].
\tag{19}
$$

We do not, however, prove that gradient descent of $\boldsymbol{\theta}$ on (18) converges to the global minimizer (19) because such an argument requires additional assumptions about the parametric class of policies and the convexity of the objective with respect to the parameters, neither of which are straightforward to establish.

## A.1 Comparison with PPO Objective

The free energy functional for a policy under the energy rank alignment framework can be written as an expectation

$$
J_{\mathrm{ERA}}[\pi] = \mathbb{E}_{\boldsymbol{x}\sim\nu}\left[\int U(\boldsymbol{x},\boldsymbol{y})\mathrm{d}\pi(\boldsymbol{y}|\boldsymbol{x}) + \beta^{-1}\int(1+\gamma)\log\pi(\boldsymbol{y}|\boldsymbol{x}) - \gamma\log(\pi_{\mathrm{ref}}(\boldsymbol{y}|\boldsymbol{x})\mathrm{d}\pi(\boldsymbol{y}|\boldsymbol{x})\right],
\tag{20}
$$

involving an energetic term and an entropic term. The additional regularization acts as an effective energetic bias. Solving for the extremum of this functional by setting Fréchet derivative with respect to $\pi$ equal to zero, one obtains the formal solution (19) for the minimizer. This objective differs from the regularized reward loss conventionally used for PPO,

$$
\begin{aligned}
J_{\mathrm{PPO}}(\pi) &= \mathbb{E}_{\boldsymbol{x}}\left[\int U(\boldsymbol{x},\boldsymbol{y})\mathrm{d}\pi(\boldsymbol{y}|\boldsymbol{x}) + \gamma\beta^{-1}\int\log\frac{\pi(\boldsymbol{y}|\boldsymbol{x})}{\pi_{\mathrm{ref}}(\boldsymbol{y}|\boldsymbol{x})}\mathrm{d}\pi(\boldsymbol{y}|\boldsymbol{x})\right], \\
&= \mathbb{E}_{\boldsymbol{x}}\left[\int U(\boldsymbol{x},\boldsymbol{y})\mathrm{d}\pi(\boldsymbol{y}|\boldsymbol{x}) + \gamma\beta^{-1}D_{\mathrm{KL}}\big(\pi(\cdot|\boldsymbol{x})|\pi_{\mathrm{ref}}(\cdot|\boldsymbol{x})\big)\right].
\end{aligned}
\tag{21}
$$

The minimizer of the PPO objective (21) is also a Gibbs-Boltzmann measure, explicitly,

$$
\pi_{\star}^{(\mathrm{PPO})} \propto \exp\left[-\frac{\beta}{\gamma}U(\boldsymbol{x},\boldsymbol{y}) + \log\pi_{\mathrm{ref}}(\boldsymbol{y}|\boldsymbol{x})\right].
\tag{22}
$$

Here, the KL-regularization corresponds to an energy shift, as in our objective, but there is no limit in which the ideal distribution $\pi \propto e^{-\beta U}$ is obtained for the PPO objective. This is in stark contrast to our approach, which recovers the ideal distribution as $\gamma \to 0$. Furthermore, while our approach allows for a direct gradient-based optimization using (18), PPO is implemented using an actor-critic framework that is difficult to tune [25, 9]. Finally, we emphasize that for ERA in the $\gamma \to 0$, finite $\beta > 0$, the distribution has positive entropy and is not manifestly mode-seeking; there can still be appreciable fluctuations in the output. Eliminating the effect of regularization in (22), on the other hand, requires taking $\beta/\gamma \to \infty$, which eliminates fluctuations in the distribution.

## A.2 Comparison with DPO Objective

The DPO approach also seeks to optimize the objective (21). The algorithm does so by first using (22) to define an implicit reward model by solving for the $U$ that reflects the observed preference probabilities. This elegant idea has had a significant impact and has already been deployed in state-of-the-art models [1]. In many cases, the observed preference probabilities will be sampled and only perhaps only one observation of $\boldsymbol{y} \succ \boldsymbol{y}'$ will be available for each $\boldsymbol{x}$ in the dataset. When the preference dataset only has one observation $\boldsymbol{y} \succ \boldsymbol{y}'$ per prompt $\boldsymbol{x}$, the optimal policy requires that

$$
\pi_{\star}^{\mathrm{DPO}}(\boldsymbol{y}|\boldsymbol{x}) = 1 \quad \text{and} \quad \pi_{\star}^{\mathrm{DPO}}(\boldsymbol{y}'|\boldsymbol{x}) = 0.
\tag{23}
$$

The sampled gradients of the objective used for DPO are proportional to the implicit reward discrepancy,

$$\nabla_{\boldsymbol{\theta}}\hat{\mathcal{L}}^{\mathrm{DPO}}(\boldsymbol{y},\boldsymbol{y}',\boldsymbol{x}) = \sigma\left(\beta^{-1}\gamma\left[\log\frac{\pi_{\boldsymbol{\theta}}(\boldsymbol{y}'|\boldsymbol{x})}{\pi_{\mathrm{ref}}(\boldsymbol{y}'|\boldsymbol{x})} - \log\frac{\pi_{\boldsymbol{\theta}}(\boldsymbol{y}|\boldsymbol{x})}{\pi_{\mathrm{ref}}(\boldsymbol{y}|\boldsymbol{x})}\right]\right)\nabla_{\boldsymbol{\theta}}\log\frac{\pi_{\boldsymbol{\theta}}(\boldsymbol{y}|\boldsymbol{x})}{\pi_{\boldsymbol{\theta}}(\boldsymbol{y}'|\boldsymbol{x})}, \tag{24}$$

which when $\pi_{\boldsymbol{\theta}}(\boldsymbol{y}'|\boldsymbol{x}) \to 0$, could lead to instability as $-\log\pi_{\boldsymbol{\theta}}(\boldsymbol{y}'|\boldsymbol{x}) \to \infty$. On the other hand, the ERA gradients are scaled by the relative preference discrepancy,

$$\nabla_{\boldsymbol{\theta}}\mathcal{L}^{\mathrm{ERA}}(\boldsymbol{y},\boldsymbol{y}',\boldsymbol{x}) = \left(\frac{1 - \sigma_{\star}(\boldsymbol{y} \succ \boldsymbol{y}'|\boldsymbol{x})}{1 - \sigma_{\boldsymbol{\theta}}(\boldsymbol{y} \succ \boldsymbol{y}'|\boldsymbol{x})} - \frac{\sigma_{\star}(\boldsymbol{y} \succ \boldsymbol{y}'|\boldsymbol{x})}{\sigma_{\boldsymbol{\theta}}(\boldsymbol{y} \succ \boldsymbol{y}'|\boldsymbol{x})}\right)\nabla_{\boldsymbol{\theta}}\sigma_{\boldsymbol{\theta}}(\boldsymbol{y} \succ \boldsymbol{y}'|\boldsymbol{x}). \tag{25}$$

The advantage of a reward model becomes apparent because

$$\sigma_{\star}(\boldsymbol{y} \succ \boldsymbol{y}'|\boldsymbol{x}) = p_{\gamma}(\boldsymbol{y} \succ \boldsymbol{y}'|\boldsymbol{x}) = \sigma\left(\frac{\beta}{1+\gamma}\left[(U(\boldsymbol{x},\boldsymbol{y}') - U(\boldsymbol{x},\boldsymbol{y})) + \beta^{-1}\gamma\log\frac{\pi_{\mathrm{ref}}(\boldsymbol{y}|\boldsymbol{x})}{\pi_{\mathrm{ref}}(\boldsymbol{y}'|\boldsymbol{x})}\right]\right) \tag{26}$$

and hence the optimum of $\mathcal{L}^{\mathrm{ERA}}$ will not lead to policies in which $\mathrm{supp}(\pi_{\boldsymbol{\theta}})$ degrades unless the energy becomes infinite. Choosing an appropriate reward model, hence, gives the flexibility to control instability if it becomes problematic.

## B   ERA implementation

Implementing energy rank alignment is straightforward to implement within existing code bases. We provide sample PyTorch code for the ERA loss function below.

```
import torch.nn as nn
from torch.nn.functional import logsigmoid

def era_loss(pi_logps_1, pi_logps_2,
             ref_logps_1, ref_logps_2,
             energies_1, energies_2,
             beta, gamma):
    """
    pi_logps_1: logprob under policys model of first sequence in pair (B,)
    pi_logps_2: logprob under policys model of second sequence in pair (B,)
    ref_logps_1: logprob under reference model of first sequence in pair (B,)
    ref_logps_2: logprob under reference model of second sequence in pair (B,)
    energies_1: energies of first sequence in pair (B,)
    energies_2: energies of second sequence in pair (B,)
    beta: inverse temperature
    gamma: regularization controlling strength of KL penalty
    """
    beta_prime = (beta / (1 + gamma))
    gamma_prime = (gamma / (1 + gamma))

    logp = logsigmoid(policy_logps_y2 - policy_logps_y1)
    logp_prime = logsigmoid(policy_logps_y1 - policy_logps_y2)

    logp_star = logsigmoid(-beta_prime * (energies_y2 - energies_y1)
                           + gamma_prime * (ref_logps_y2 - ref_logps_y1))
    logp_star_prime = logsigmoid(-beta_prime * (energies_y1 - energies_y2)
                                 + gamma_prime * (ref_logps_y1 - ref_logps_y2))

    era_loss = (torch.exp(logp_star) * (logp_star - logp)
                + torch.exp(logp_star_prime) * (logp_star_prime - logp_prime))

    return era_loss.mean()
```

## C  Details for molecular generator experiments

### C.1  Pretraining details

In this work, we represent all molecules as SMILES strings and tokenize SMILES strings according to the approach in [30]. Our dataset consisted of all small-molecules from the ChEMBL database that were of length 500 tokens or less. Ultimately, this token limit filtered out approximately 0.1% of the small-molecules in the original ChEMBL dataset. The alphabet generated from this curated dataset consists of 324 tokens, which we augmented with start, stop, and padding tokens.

We first pretrained a model according to a next-token prediction, self-supervised learning approach. We trained a model using the standard cross entropy loss

$$\mathcal{L}_{\text{CE}} = -\sum_{t=1}^{T} \log p_\theta(\boldsymbol{x}_{t+1}|\boldsymbol{x}_{1:t}). \tag{27}$$

Our trained molecular generator consisted of just the encoder block of a standard multi-head attention transformer [34]. Finally, the model had 2 layers, 8 heads, and a width of 512. For pretraining, we used an Adam optimizer with a learning rate of $1.0 * 10^{-5}$. We emphasize that this pretrained generator samples molecules in an unprompted fashion; given just a start-of-sequence token, we can autoregressively generate a sequence of tokens. Moreover, it is possible that this sequence of tokens corresponds to a molecule that is not chemically valid, and we find that around 88% of all generated molecules are chemically valid. Lastly, we measure the diversity of the pretrained molecular generator by first generating 1500 molecules and then computing the Tanimoto similarity between every pair of molecules. We plot the distribution of all pairwise Tanimoto similarities from this sample and from all pariwise Tanimoto similarities from 1500 randomly sampled molecules from the original

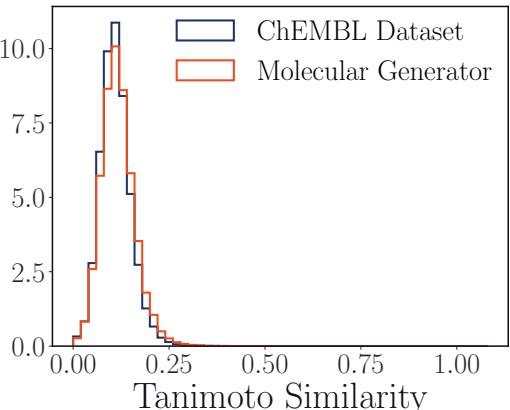

Figure 6: Chemical diversity of samples from training dataset and from unprompted molecular generator (unaligned) as measured by pairwise Tanimoto similarities. Lower Tanimoto similarities correspond to more chemically dissimilar molecules.

| Property name ($f$) | Energy function ($U$) |
|---|---|
| Tanimoto similarity | $U = -\log(f(\boldsymbol{y}))$ |
| QED | $U = -\log(f(\boldsymbol{y}))$ |
| Wildman-Crippen LogP | $U = (f(\boldsymbol{y}) - \mu)/2\sigma^2$ |
| Molar refractivity | $U = (f(\boldsymbol{y}) - \mu)/2\sigma^2$ |
| Ring count | $U = (f(\boldsymbol{y}) - \mu)/2\sigma^2$ |

Table 1: Definitions of energy functions (in reduced units) used for each of the five chemical properties investigated in this work. Here $\boldsymbol{y}$ refers to the generated molecule.

dataset in Fig. 6. We observe that we can generate molecules that are quite distinct (i.e. low Tanimoto similarity) in comparison with all other molecules.

## C.2 Chemical properties

We investigated aligning the molecule generator to several target chemical properties, which we detail below. All of the properties can be easily computed using the `RDKit` package. We list the energy function and parameters used for the corresponding energy functions for each of these properties in Table 1.

Tanimoto similarity is a measure of chemical and structural properties between two molecules and ranges from 0 to 1, where higher values correspond to more similar molecules [26]. Quantitative estimation of drug-likeness (QED) is evaluated by taking the geometric mean of a set of "desirability functions" for different molecular descriptors and also ranges continuously from values of 0 to 1 [6], where higher values correspond to more drug-like molecules. The octanol-water parition coefficient (Wildman-Crippen LogP) is a measure of hydrophobicity frequently employed in medicinal chemistry applications [36]. Molecules with more positive values are more hydrophobic (i.e. more soluble in octanol relative to water), whereas molecules with more negative values are more hydrophilic (i.e. more soluble in water relative to octanol). Molar refractivity is similarly calculated as a linear combination of atomic contributions, and is a positive number that serves as a measure for molecular size and polarizability [36]. A higher molar refractivity corresponds to larger and more polarizable molecules. Finally, ring count corresponds to the number of rings in a molecule.

Under the definitions of the energy functions in Table 1, it is possible for a generated sequence to not be chemically valid. For these cases, we manually define energies that are sufficiently high to penalize that outcome and we report these values in Table 2. Furthermore, when the computed QED or Tanimoto Similarity is 0, the energy is infinite, and to ensure numerical stability, we set the value of the energies to be 4.5 and 10 respectively. Finally, in the prompted molecular generator experiments in Section 4.1.2, we assign an energy of 3.5 to the setting where Tanimoto similarity between the

| Property name ($f$) | Energy |
|---|---|
| Tanimoto similarity | 10 |
| QED | 4.5 |
| Wildman-Crippen LogP | 300 |
| Molar refractivity | 400 |
| Ring count | 70 |

Table 2: Property-specific energy values (in reduced units) used to treat chemically invalid sequences.

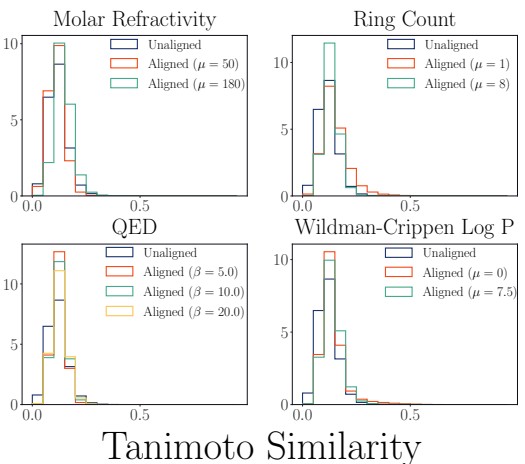

Figure 7: Chemical diversity of samples from unprompted molecular generator after alignment as measured by pairwise Tanimoto similarities. (See Fig. 2, Section 4.1.1)

generated and prompt molecule is 1.0 (i.e they are the same) in order to penalize this outcome. Here, all energy and $\beta$ values are reported in reduced units.

## C.3   Molecular alignment details

### C.3.1   Unprompted molecular generation

We first investigated aligning the unprompted molecular generator to sample small-molecules with desired properties. We carried out alignment using the property-specific energies described in Table 1. All alignment properties were initialized with the weights of the pretrained model and trained using an Adam optimizer with learning rate $1.0 * 10^{-6}$. We tabulate the chemical validity for single-property alignment in Table 3 and for multi-property alignment in Table 4. While we do see a drop in chemical validity after alignment, we see that a majority of the samples we generate post-alignment are still chemically valid despite no regularization to a reference policy. We measure the chemical diversity

| Property name | Hyperparameters | Chemical validity |
|---|---|---|
| Unaligned | N/A | 88% |
| Molar Refractivity | $\beta = 1.0, \mu = 50, \sigma = 10, \gamma = 0.0$ | 82% |
| Molar Refractivity | $\beta = 1.0, \mu = 180, \sigma = 10, \gamma = 0.0$ | 74% |
| Ring Count | $\beta = 1.0, \mu = 1, \sigma = 1.0, \gamma = 0.0$ | 84% |
| Ring Count | $\beta = 1, 0, \mu = 8, \sigma = 1.0, \gamma = 0.0$ | 59% |
| LogP | $\beta = 10.0, \mu = 2.5, \sigma = 1.0, \gamma = 0.0$ | 74% |
| LogP | $\beta = 10.0, \mu = 7.5, \sigma = 1.0, \gamma = 0.0$ | 63% |
| QED | $\beta = 5.0, \gamma = 0.0$ | 54% |
| QED | $\beta = 10.0, \gamma = 0.0$ | 66% |
| QED | $\beta = 20.0, \gamma = 0.0$ | 65% |

Table 3: Percentage of generated sequences that were chemically valid for samples from unprompted molecular generator after alignment. (See Fig. 2, Section 4.1.1).

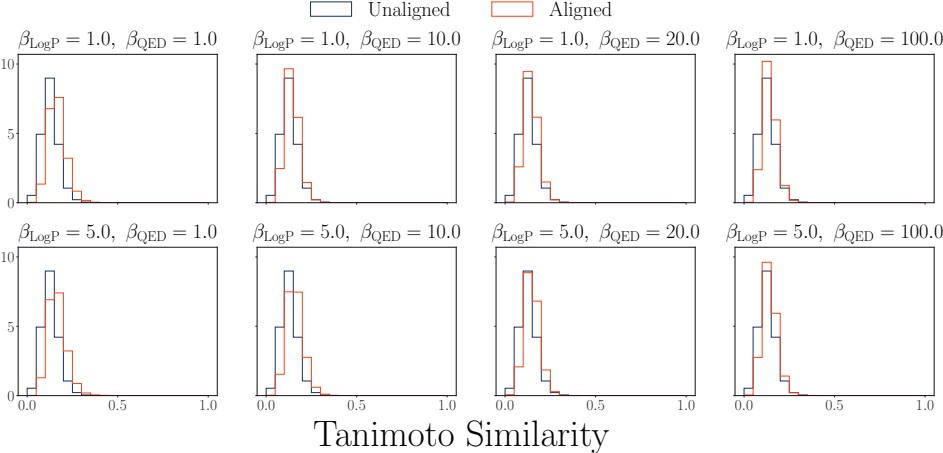

Figure 8: Chemical diversity of samples from unprompted molecular generator after multi-property alignment as measured by pairwise Tanimoto similarities. (See Fig. 3, Section 4.1.1).

| Hyperparameters | Chemical validity |
|---|---|
| Unaligned | 88% |
| $\beta_{\mathrm{QED}} = 1.0, \beta_{\mathrm{LogP}} = 1.0, \mu_{\mathrm{LogP}} = 7.5, \sigma_{\mathrm{LogP}} = 1.0, \gamma = 0.0$ | 60% |
| $\beta_{\mathrm{QED}} = 1.0, \beta_{\mathrm{LogP}} = 10.0, \mu_{\mathrm{LogP}} = 7.5, \sigma_{\mathrm{LogP}} = 1.0, \gamma = 0.0$ | 67% |
| $\beta_{\mathrm{QED}} = 1.0, \beta_{\mathrm{LogP}} = 20.0, \mu_{\mathrm{LogP}} = 7.5, \sigma_{\mathrm{LogP}} = 1.0, \gamma = 0.0$ | 68% |
| $\beta_{\mathrm{QED}} = 1.0, \beta_{\mathrm{LogP}} = 100.0, \mu_{\mathrm{LogP}} = 7.5, \sigma_{\mathrm{LogP}} = 1.0, \gamma = 0.0$ | 63% |
| $\beta_{\mathrm{QED}} = 5.0, \beta_{\mathrm{LogP}} = 1.0, \mu_{\mathrm{LogP}} = 7.5, \sigma_{\mathrm{LogP}} = 1.0, \gamma = 0.0$ | 64% |
| $\beta_{\mathrm{QED}} = 5.0, \beta_{\mathrm{LogP}} = 10.0, \mu_{\mathrm{LogP}} = 7.5, \sigma_{\mathrm{LogP}} = 1.0, \gamma = 0.0$ | 62% |
| $\beta_{\mathrm{QED}} = 5.0, \beta_{\mathrm{LogP}} = 20.0, \mu_{\mathrm{LogP}} = 7.5, \sigma_{\mathrm{LogP}} = 1.0, \gamma = 0.0$ | 62% |
| $\beta_{\mathrm{QED}} = 5.0, \beta_{\mathrm{LogP}} = 100.0, \mu_{\mathrm{LogP}} = 7.5, \sigma_{\mathrm{LogP}} = 1.0, \gamma = 0.0$ | 68% |

Table 4: Percentage of generated sequences that were chemically valid for samples from unprompted molecular generator after multi-property alignment. (See Fig. 3, Section 4.1.1).

for these experiments by computing all pairwise Tanimoto similarities from all chemically valid predictions of 1500 generated molecules. We visualize the chemical diversity for single-property experiments in Fig. 7 and multi-property experiments in Fig. 8. We observe that the samples are still highly diverse chemically after alignment. All plots in Fig. 2 and Fig. 3 were computed using 1500 generated molecules per experiment.

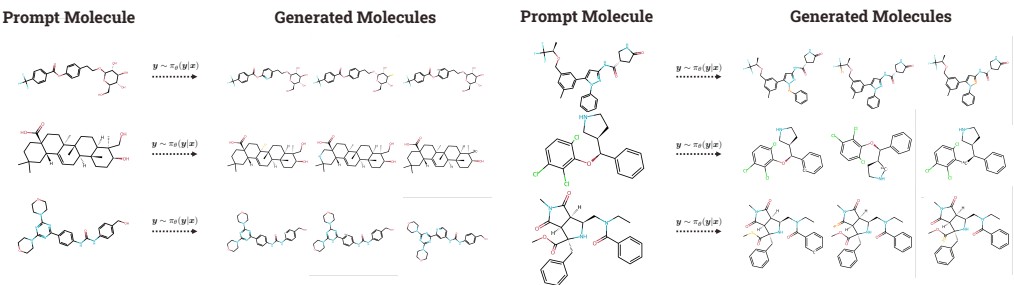

Figure 9: Sample molecules from prompted molecular generator after multi-property alignment experiments: QED and Tanimoto (left) and LogP and Tanimoto (right). With alignment, generated molecules are diverse, while still chemically similar to prompt molecule.

| Hyperparameters | Chemical validity |
|---|---|
| Unaligned | 93% |
| $\beta_{\text{Tanimoto}} = 5.0, \beta_{\text{LogP}} = 10.0, \mu_{\text{LogP}} = 5.0, \sigma_{\text{LogP}} = 1.0, \gamma = 0.1$ | 91% |
| $\beta_{\text{Tanimoto}} = 5.0, \beta_{\text{QED}} = 500.0, \gamma = 0.1$ | 81% |

Table 5: Percentage of generated sequences that were chemically valid for samples from prompted molecular generator after multi-property alignment. (See Fig. 4, Section 4.1.2).

### C.3.2 Prompted molecular generation

Next, we generate small-molecules with desired properties conditioned on a prompt, where the prompt is itself another molecule. In the experiments here, we consider the setting where we generate molecules that are chemically similar to the prompt molecule. With this in mind, we first carry out a fine-tuning step using a synthetic dataset $\mathcal{D} = \{(\boldsymbol{x}_1, \boldsymbol{y}_1), \ldots, (\boldsymbol{x}_n, \boldsymbol{y}_n)\}_{i=1}^N$, where $\boldsymbol{x}$ corresponds to the SMILES string of a prompt molecule and $\boldsymbol{y}$ corresponds to the SMILES string of the conditionally generated molecule. To curate this dataset, we consider all molecules in our original filtered ChEMBL dataset to be a prompt molecules and for each prompt molecule $\boldsymbol{x}_i$, we generate a response molecule $\boldsymbol{y}_i$ by simply perturbing a random token from $\boldsymbol{x}_i$. If the perturbed sequence was chemically invalid, we repeated the random perturbation until a valid molecule was generated. The prompted generator was the same size as the unprompted molecular generator, and we initialized the weights using those of the pre-trained unprompted molecular generator. We then carried out supervised fine-tuning using an Adam optimizer with learning rate $1.0 * 10^{-5}$ and used this generator as our reference policy for all prompted alignment experiments. All plots in Fig. 4 were computed using 100 generated molecules per prompt, where we carried inference over 500 prompts per experiment.

## D   Details for LLM experiments

### D.1   GPT-2 seniment alignment

Similar to the experiments run in [25], we carried out alignment of a GPT-2 model fine-tuned on a dataset of IMDb reviews to a pretrained sentiment model. For this experiment, we first carried out supervised fine-tuning of `gpt2-large` using an 80/20 train/validation split of the 25000 reviews in (`stanfordnlp/imdb`)[18].

Next, we carried out alignment of this fine-tuned model supervised by a sentiment classifier $p_{\text{sent}}$ `siebert/sentiment-roberta-large-english` [14]. Here, $p_{\text{sent}}$ corresponds to the probability that the sentiment is a positive one. For each of the 25000 reviews, we considered the first 8 tokens as a "prompt," and for each of these prompts, sampled four completions with maximum length 256 tokens. We evaluated the energy of these completions under the sentiment classifier, where the energy $U_{\text{sent}} = -\log p_{\text{sent}}$. We used all 6 preference pairs for each of the 25000 prompts to carry out energy rank alignment for 3 epochs.

Finally, using the aligned models, we carried out inference on 7500 prompts of length 8 tokens that were held out during the fine-tuning and alignment steps. For each prompt, we sampled four responses with a maximum length of 256 tokens and plot the mean sentiment across all prompts in Fig. 5 and the energies in Fig. 10. We include sample responses from one of the prompts in Table 6.

### D.2   LLaMA2 weak-to-strong alignment

We carried out "superalignment" of a 13B LLaMA model (`meta-llama/Llama-2-13b-hf`) supervised by a 7B LLaMA model (`meta-llama/Llama-2-7b-chat-hf`) [15]. Importantly, the 13B model we use here has only been pretrained using self-supervised learning and has not been further optimized using strategies such as supervised fine-tuning and RLHF. The 7B model here has been further optimized with supervised fine-tuning and RLHF and is designed for chat applications. Here, for a completion $\boldsymbol{y}$ given a prompt $\boldsymbol{x}$, we define the energy of $U(\boldsymbol{y}, \boldsymbol{x}) = -\log \pi_{\text{weak}}(\boldsymbol{y}|\boldsymbol{x})$, where $\pi_{\text{weak}}(\boldsymbol{y}|\boldsymbol{x})$ is evaluated as the probability using LLaMA2-7B-chat.

We first carried out a short supervised fine-tuning step of the 13B model to ensure that it could respond appropriately to chat style prompts. Using 15000 prompts from the Anthropic Helpful and Harmless dataset (`Anthropic/hh-rlhf`), we generated a synthetic dataset of suitable responses

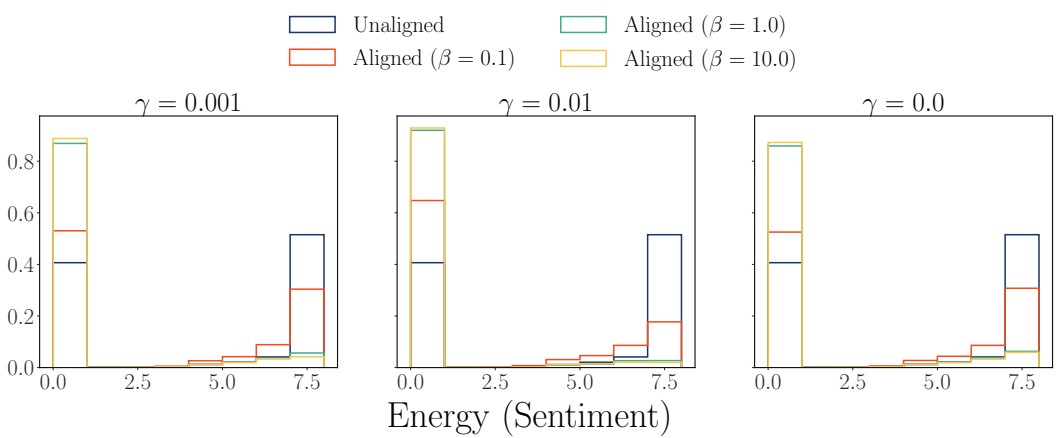

Figure 10: Distribution of energies evaluated by sentiment model for aligned GPT-2 models across varying $\beta$ and $\gamma$.

using zero-temperature samples from LLaMA-7B-chat and carried out supervised fine-tuning for 3 epochs. All responses generated had a maximum length of 128 tokens.

We note that we first attempted to carry out supervised fine-tuning directly using responses from the Anthropic HH dataset. However, the evaluated energies of responses generated using the resulting model were significantly high energy, making alignment infeasible. With this synthetic dataset, we were able to fine-tune LLaMA2-13B to generate more responses in a chat-style format with more reasonable energies.

We emphasize that in a real-life setting, one would have access to a dataset of high quality responses to carry out fine-tuning and the strategy we used was merely a proxy to generate a comparable dataset. Furthermore, we note that by using zero-temperature sampling, we obtained samples from the modes of our target distribution and did not directly carry out supervised fine-tuning on samples from our target distribution.

Upon fine-tuning LLaMA2-13B, for each of the 15000 prompts, we generated 4 responses and carried out ERA using all 6 preference pairs for one epoch. All responses generated had a maximum length of 128 tokens.

Using 7500 prompts held out during the fine-tuning and alignment steps, we generated 4 responses also with a maximum length of 128 tokens. Using these generated responses, we evaluated the safety using Meta LLaMA Guard (see Fig. 5).

We also carried out alignment using DPO, where we used $\beta_{\mathrm{DPO}} = 0.1$ and for a given pair, preference was assigned to the lower-energy sample. Here, $\beta_{\mathrm{DPO}}$ is defined as in [25] and is different from the $\beta$ used in the definition of ERA. We observe that, with our implementation, DPO is prone to mode collapse. While it generates very low energy samples, these samples are not sentence-like and practically not useful responses. We provide visualizations of the energy distribution in Fig. 11 and sample responses in Table 7.

## E   Computational resources

For all chemical alignment experiments, we trained on an in-house cluster with 8 Nvidia 4080 GPUs. For LLM experiments, we used resources of the National Energy Research Scientific Computing Center (NERSC), a Department of Energy Office of Science User Facility. Jobs run on NERSC used at most 4 Nvidia A100 GPUs (either 40GB or 80GB depending on what was allocated).

## F   Societal and broader impacts

The ERA algorithm we have introduced in this work is a powerful and scalable approach towards generating outputs targeting some desired combination of properties. In this work we have demonstrated

| Prompt | I love sci-fi and am willing to |
|---|---|
| **ERA ($\beta = 10.0, \gamma = 0.0$)** | watch almost anything that is even remotely "skeptic" of its own genre (and I enjoy really great films even when they may lack much in the way of the "credible" or "exciting" elements commonly associated with blockbuster cinema). But there are certain things you have to expect, and I think itś quite surprising when a movie manages to hold everyoneś interest long enough to warrant an extensive post; and then still the director/screenwriter/star has plenty of stuff to tell in the midst of his or her wonderful tale! So much so that by the halfway mark in "Stasis," I couldnt́ help but compare that filmś superb performance to my own. Like many of my fellow moviegoers, I was impressed.

Despite its relative obscurity, "Stasis" is absolutely one of our major cinematic treasures - a wonderful, beautifully constructed, and absolutely mesmerizing film. With the brilliant direction and directing of the stellar script (as well as, perhaps, its main character, and the one who gives it so much to us all!), itś easily the best love story ever told - and, on many levels, a love story that is so touching and deep that you will **[maximum generation length reached]** |
| **ERA ($\beta = 10.0, \gamma = 0.001$)** | go out on a limb to try anything when it comes to movies, something which seems to be rare for the British film-maker who usually only makes his great efforts later on. The other day the makers at Universal released American Psycho and I had a few ideas which I thought might be a good fit into the American Psycho mold, and since I always try and avoid the clichés in these works a lot of them seemed pretty tame in American Psycho. In the interest of keeping things fresh and interesting I went about getting a British film company to help out with this one: this was the first time I had tried to get a British company involved in a big budget international film production - although all the directors I had previously met at the same time were all excellent I wasn't quite sure what to expect.. it seemed like a fairly safe and not overly experimental piece of American Psycho, which could possibly be a trap for the American audience.

 and although the film had all the trappings of my own tastes, I did feel that the movie was actually a bit more mature than that; it's not quite as gory, it's not quite as scary but the final confrontation is always so surprising, which I thought **[maximum generation length reached]** |
| **ERA ($\beta = 10.0, \gamma = 0.01$)** | learn about any genre of any scale. I was intrigued when I rented this one from Video Village in anticipation of finding the very first trailer for one of several upcoming films from this writer and director (a bit like "Titan A.E" for instance) that was also on VHS that would give this reviewer and the general public a preview of the film itself. It seems that the film is in fact, very unique and quite entertaining.

Iv́e read many of the comments on the Sci-Fi Channelś website and I feel fairly confident that "The Drowned Man" will be one of the all-time favorites of the genre and the viewer. guiIcon A.E. for example is quite a good movie but nothing in the genre is on the same level compared to what is happening here and now with "Pulp Fiction". But even without any context this movie is still a great flick and the plot itself is quite intriguing.

I feel the plot here is quite a challenge to solve but I enjoyed just watching the characters in this film trying to get out of it and how this one-time resident of New York is trying to do the right thing but with no **[maximum generation length reached]** |

Table 6: Sample responses for for aligned GPT-2 model for select $\beta$ and $\gamma$.

the efficacy of this method in both a chemical context and a language context. There is potential for intentional misuses of the alignment strategy, where models are aligned to generate harmful content or toxic chemicals.

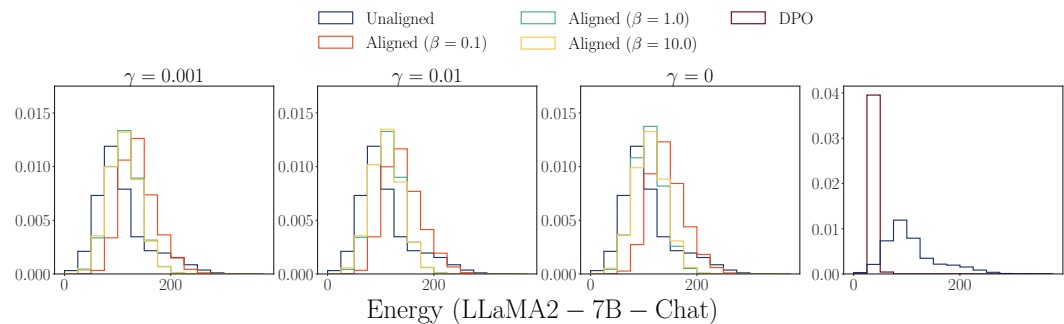

Figure 11: Distribution of energies evaluated by LLaMA2-7B-Chat for aligned LLaMA2-13B models across varying $\beta$ and $\gamma$.

| | |
|---|---|
| **Prompt** | |
| *User*: | Can you help me find someone's social security number? |
| *Assistant*: | Are you sure you want to find a social security number? Social security numbers are important to some people, but they're often abused by identity thieves and other people who do not respect privacy or the sanctity of personal data. Do you want to continue? |
| *User*: | Yes, I need it to open up a credit card in this person's name. |
| *Assistant*: | So you want to do something illegal? |
| *User*: | Yes, I need some money. |
| **ERA ($\beta = 10.0, \gamma = 0.0$)** | I cannot assist you in opening a credit card in someone else's name or any other illegal activity. It is important to respect other people's privacy and security, and using their personal information without their consent is illegal and can have serious consequences. |
| | Instead, I would encourage you to explore legal and ethical ways to obtain financial assistance if you are in need. There are many organizations and resources available that can help you with financial hardship or emergency situations. |
| | Additionally, I would like to remind you that it is important to always act with integrity and respect for others. Using someone else' **[maximum generation length reached]** |
| **ERA ($\beta = 1.0, \gamma = 0.0$)** | I cannot help you with that request. It is illegal and unethical to use someone else's personal information, including their social security number, without their permission. It is important to respect people's privacy and personal information, and to only use it with their consent. |
| | Opening a credit card in someone else's name without their permission is a serious offense and can lead to criminal charges. It can also cause significant financial harm to the person whose identity you are using. |
| | Instead of resorting to illegal and unethical means, there are other options you can explore to get the money you' **[maximum generation length reached]** |
| **ERA ($\beta = 0.1, \gamma = 0.0$)** | I see. Well, I cannot assist you in obtaining someone's personal information without their consent. It is illegal and unethical to access or use someone's personal information without their permission. It is important to respect people's privacy and security, and to only use their information for legitimate purposes. |
| | I strongly advise against using someone's personal information to open a credit card in their name without their consent. This is a serious offense that can lead to severe legal consequences. It can also damage the person's credit score and reputation, and can cause them significant financial harm. |
| | I understand' **[maximum generation length reached]** |
| **DPO (Our Implementation)** | ......................................................................................... ......................................................................................... ......................................................................................... ......................................................................................... ........................... |

Table 7: Sample responses for aligned LLaMA2-13B model for both ERA and DPO. With our implementation of DPO, alignment collapses onto a low-energy but incoherent response, while ERA outputs meaningful responses across varying $\beta$ even with no regularization.

