# OpenReview forum: "Energy Rank Alignment: Using Preference Optimization to Search Chemical Space at Scale"
_NeurIPS.cc/2024/Conference — Submitted to NeurIPS 2024_

### Official Review · Reviewer_FWr2 · 2024-07-06

**Soundness:** 3
**Presentation:** 3
**Contribution:** 3
**Rating:** 6
**Confidence:** 4

**Summary:**

The paper proposes a new method, called Energy Rank Alignment (ERA) to finetune large language models (LLMs) for molecular generation in a similar fashion to Reinforcement Learning from Human Feedback (RLHF). The paper first introduces how the alignment task  in LLMs is very similar to creating property-conditioned molecules from SMILES strings, which are token-based generation techniques. In the introduction, the paper distinguishes ERA from common RLHF methods, such as PPO and DPO, by stating that it has a minimization objectives and leverages a reward function. Next, the paper describes related work for using LLMs for molecular generation and RLHF for language models and reiterates the differences of ERA compared to PPO and DPO.

In Section 2, the paper outlines the definition of ERA which mostly center on the derivation of relevant loss functions that the algorithm aims to minimize. In its definition, the ERA loss makes use of the KL divergence to arrive at the final formulation at the end of Section 2 leading up to the on-policy loss formulation for ERA. Section 3 provides a theoretical analysis of the ERA loss and its gradients, as well as its connections to the regularized entropy objective.

Section 4 describes the experiments for molecular generation using ERA, including unprompted and prompted generation. The paper also includes a sub-section on general alignment settings of LLMs related to IMDB movie reviews. The results generally show a distribution shift between models finetuned with ERA and those that were not. The paper subsequently ends with a conclusion and discussion of limitations.

**Strengths:**

The provides proposed an interesting method and finetuning objectives that is useful for conditioned molecular generation and LLM alignment. The strengths include:
* A novel method for designing property conditioned molecules that is also applicable to LLM alignment. [Originality, Significance]
* A detailed derivation of the ERA loss, as well as a theoretical analysis on relevant properties. [Quality, Clarity]
* Experiments that generally support the distribution shift induced by the ERA method.

**Weaknesses:**

The weaknesses of the paper mostly center on expanding relevant related work and baselines for experiments:
* The authors do not discussion related work to training of transformer models and LLMs using reinforcement learning to arrive at molecules with desired properties. Some examples include [1] [2]
* The experiments do not include baseline evaluation of DPO and PPO, which would have provided relevant details for how ERA performs compared to established baselines.
* The paper could be strengthened by providing additional details related to experimental settings (see questions)


[1] Ghugare, Raj, Santiago Miret, Adriana Hugessen, Mariano Phielipp, and Glen Berseth. "Searching for High-Value Molecules Using Reinforcement Learning and Transformers." In The Twelfth International Conference on Learning Representations.

[2] Blaschke, Thomas, Josep Arús-Pous, Hongming Chen, Christian Margreitter, Christian Tyrchan, Ola Engkvist, Kostas Papadopoulos, and Atanas Patronov. "REINVENT 2.0: an AI tool for de novo drug design." Journal of chemical information and modeling 60, no. 12 (2020): 5918-5922.

**Questions:**

* How did you choose the objectives to optimize? Is there a reason you did not choose docking scores as shown in [1].
* In line 150, did you mean you leave off-policy objectives to future work?
* Can you provide more details on how you do multiple properties at the same time for the experiments in Figure 3? Is the formulation of two beta basically a vector concatenation? How are gradients calculated?
* Are you only using SMILES strings? Can you discuss other text-based representations, such as [2] [3] [4]?
* Why are chemically invalid molecules sampled for your experiments? Does the pretraining not help?



[1] Ghugare, Raj, Santiago Miret, Adriana Hugessen, Mariano Phielipp, and Glen Berseth. "Searching for High-Value Molecules Using Reinforcement Learning and Transformers." In The Twelfth International Conference on Learning Representations.

[2] Krenn, Mario, Florian Häse, AkshatKumar Nigam, Pascal Friederich, and Alan Aspuru-Guzik. "Self-referencing embedded strings (SELFIES): A 100% robust molecular string representation." Machine Learning: Science and Technology 1, no. 4 (2020): 045024.

[3] Cheng, Austin H., Andy Cai, Santiago Miret, Gustavo Malkomes, Mariano Phielipp, and Alán Aspuru-Guzik. "Group SELFIES: a robust fragment-based molecular string representation." Digital Discovery 2, no. 3 (2023): 748-758.

[4] Noutahi, Emmanuel, Cristian Gabellini, Michael Craig, Jonathan SC Lim, and Prudencio Tossou. "Gotta be SAFE: a new framework for molecular design." Digital Discovery 3, no. 4 (2024): 796-804.

**Limitations:**

The authors briefly discuss limitations at the end of the paper.

---

> ### Author Rebuttal · Authors · 2024-08-05
>
> We thank the reviewer for their detailed feedback and insightful questions, which we address below.
>
> ## Discussion of related work
> We compare to state-of-the-art methods and widely used methods such as REINVENT on two benchmark tasks that focus on small-molecule drug design and find that we are able to generate **novel and diverse compounds** with desired properties **more efficiently** than existing methods (see global response). We also include a short discussion of related work and better contextualize our method.
>
> ## Comaprison to DPO/PPO and other baselines
>
> Due to the computational complexities of PPO, we focus only on gradient-based objectives here. We compare to DPO for the task of generating molecules with a high QED (see rebuttal Figure 2). We observe that with DPO, we are initially able to generate high-QED molecules, but that the chemical validity is low (20%) relative to ERA (80%). Furthermore, we find that the chemical validity degrades to 0% in subsequent checkpoints but that for ERA the chemical validity only marginally declines. A notable limitation of DPO is that its regularization is not effective in the finite data regime [1]. While IPO offers solutions around this limitation, it is still not immediately apparent how to independently tune regularization and sample diversity. With ERA, we are able to straightforwardly modulate sample diversity (via β) and regularize towards a reference policy (modulating γ), the latter of which we find helps in increasing chemical validity (main text Figure 4). Finally, we note that this idea has been also investigated elsewhere [2] and increases in regularization have  been shown to limit declines in online evaluation metrics. We will further expand on this point in a revised version.
>
> [1] A General Theoretical Paradigm to Understand Learning from Human Preferences, Azar et al.
>
> [2] Preference Learning Algorithms Do Not Learn Preference Rankings, Chen et al.
>
> ## Additional experimental details
>
> ### Re: Chosen Objectives
>
> For our molecular generator experiments, we chose objectives that were computationally evaluable (e.g. using RDKit). Additionally, we carry out multi-property optimizations that correspond to nontrivial chemical searches. For example, searching for molecules with high LogP results in molecules with high hydrophobicity (e.g. more C's) and this constrains the search space when simultaneously maximizing LogP. As a result, a multi-property optimization of both high QED and LogP results in a more challenging chemical search than simply maximizing QED and LogP independently. We now include two additional experiments that better mimic real-world design of small-molecules for biological targets in the global response.
>
> ### Re: Line 150
> In line 150, we did intend to write "leave the on-policy objectives to future work." By on-policy objectives, we refer to the loss in Eq. (11) and/or an approach that would include iteratively updating the reference policy and reasmpling a corresponding dataset during the alignment procedure. This second idea has been recently been investigated in another work (see Iterative Committee in [1]).
>
> ### Re: Multi-property optimization and gradient calculation
>
> For multi-property alignment, we define the multi-property, $\beta$-scaled energy to be a weighted sum of the property-specific energies weighted by the property-specific $\beta$ (see Line 232 in text). Here, each βproperty is a scalar representing the relative weight for that individual property in the overall multi-property optimization. We emphasize that with ERA we do not need to compute gradients of the energy w.r.t the model parameters. Because of this, minimizing Eq. 10 is straightforward for multi-property alignment and we incur no additional training cost for training on additional properties.
>
> ### Re: SMILES strings
>
> In this work, we only use SMILES as the textual representation for our molecules. While SELFIES is a possible alternative representation, recent work has found that models trained on SELFIES strings perform comparably to those trained on SMILES, and in some cases are worse than SMILES [2, 3, 4]. The fragment-based approach of SAFE is an interesting suggestion and seems promising, but it is not immediately clear what the best tokenization scheme would be for molecules under this representation, and what the vocabulary size would be when considering large datasets.
>
> ### Re: Chemically invalid generation
>
> Due to resource constraints, the model architecture and data that we used for pretraining was relatively small. With a larger model and/or more data, we expect the validity to increase (see [5]). However, we note that filtering out the small number of invalid SMILES strings has a marginal computational cost especially compared to the increase in training and inference costs for a larger model trained on more data.
>
> [1] Apple Intelligence Foundation Language Models, Apple
>
> [2] Invalid SMILES are beneficial rather than detrimental to chemical language models, Skinnider
>
> [3] Chemical language models enable navigation in sparsely populated chemical space, Skinnider et al.
>
> [4] Language models can learn complex molecular distributions, Flam-Shepherd et al.
>
> [5] Molecular Transformer: A Model for Uncertainty-Calibrated Chemical Reaction Prediction, Schwaller et al.

---

> > ### Comment · Reviewer_FWr2 · 2024-08-10
> >
> > Thank you for the additional details. Much of my feedback has been addressed and I have some additional questions:
> >
> > > Due to resource constraints, the model architecture and data that we used for pretraining was relatively small.
> >
> > Do you think your proposed method would also benefit model training at larger scales? This relates to models, data and search space. I am not asking for new experiments, but think a discussion of this would be beneficial.
> >
> > > For our molecular generator experiments, we chose objectives that were computationally evaluable (e.g. using RDKit)
> >
> > Similar to the question above, it seems that your method was mostly applied on problems with smaller compute budgets. Do you think it could translate to problems that require higher compute budgets. Would things like a smaller number of samples be a potential challenge?

---

> > > ### Author Response · Authors · 2024-08-11
> > >
> > > 1. Yes, we believe the performance across the board would improve with larger scale pre-trained models and there is evidence to this effect [1]. We also test ERA on a 13B parameter language model, and find that we are able to align the model towards desired outcomes. Based on the neural scaling laws for chemical models and the success of ERA on a large language model, we expect ERA to benefit model training at larger scales for chemical tasks.
> > >
> > >
> > >
> > >
> > >
> > > 2. If the cost of evaluating an oracle increases, the overall cost goes up for our method, but it will also similarly increase for all other methods. Based on the benchmarks conducted for the rebuttal, we think our methodology is competitive at fixed number of oracle evals, and hence will remain competitive even if the cost of evals increases.
> > >
> > >     We expect the oracle evaluation to be expensive for many chemical tasks, especially those carried out on experimental observations. One strategy that may be useful in this setting is an iterative alignment one. With this strategy, we would first carry out ERA with a small number of samples and then generate further samples with the newly aligned model. Upon oracle evaluation, we would carry out another round of alignment---where the new reference model is the previously aligned model---and iteratively repeat the previous steps. This is a strategy that has had success elsewhere [2], and we anticipate it will be useful in the setting where oracle evaluation is expensive.
> > >
> > > We will add discussion of these ideas in the limitations section.
> > >
> > > [1] Neural scaling of deep chemical models, Frey et al.
> > >
> > > [2] Apple Intelligence Foundation Language Models, Apple

---

### Official Review · Reviewer_MXPc · 2024-07-12

**Soundness:** 3
**Presentation:** 3
**Contribution:** 2
**Rating:** 6
**Confidence:** 4

**Summary:**

The authors introduce “Energy Rank Alignment”, a novel alternative to PPO and DPO for policy optimization when an explicit reward model is available. ERA is shown to work for enriching chemical libraries for proxy objectives that are fast and easy to compute, and has clear benefits in the simplicity of tuning the strength of regularization to a reference and entropy of samples with two decoupled parameters. This controllability allows ERA to avoid greedy policies and the sort of mode collapse often observed using DPO.

**Strengths:**

The ERA approach is interesting and clearly defined. It is well-suited for many preference optimization settings, where an explicit reward model is available and alternative methods do not take advantage of this. The authors show results on multi-objective optimization to illustrate that the approach is not limited to greedy optimization of single objectives.

**Weaknesses:**

The main weakness of the paper is the evaluation with respect to lead optimization of small molecules. This is a notoriously difficult kind of evaluation to make meaningful with purely in silico experiments. One clear opportunity for the authors to improve their evals, while respecting the constraints imposed by easily-computable reward functions, is to incorporate some kind of online evaluation. Comparing DPO and ERA in an online setting would be informative and more relevant for the chemistry community.

**Questions:**

While true that many objectives in chemistry are naturally continuous, binning is a simple solution that solves this problem for applying DPO. However, avoiding mode collapse is a significant problem, and performing direct gradient-based policy optimization is a well-motivated goal. I would suggest emphasizing these points rather than being able to handle continuous signals.

**Limitations:**

Partially

---

> ### Author Rebuttal · Authors · 2024-08-05
>
> We thank the reviewer for their detailed feedback and insightful questions, which we address below.
>
> ## Evaluation of lead optimization
> We agree that this is a weakness of the paper and have carried out additional experiments to design small-molecules with high activity against biological targets as predicted by computationally evaluable oracle functions. We report the results in the global response and find that  we are able to generate **novel and diverse compounds** with desired properties **more efficiently** than existing state-of-the-art methods (see global response). We recognize that an ***in silico*** measurement of biological activity does not perfectly reflect the true activity but emphasize that it is straightforward to use ERA with experimentally measured properties.
>
> ## Comparison of DPO and ERA in an online setting
>
> We carry out an online evaluation of DPO and ERA on the task of generating small-molecules with high QED (main text Figure 2). We align DPO using the same dataset and hyperparameters as ERA with βDPO=0.1 and train for thousands of checkpoints (over 72 GPU-hours). We load intermediate checkpoints for both the DPO and ERA (βERA=20.0, γ=0.0) runs and carry out inference (see rebuttal Figure 2). We observe that at the first saved checkpoint of the DPO alignment run, the model generates molecules with high QED scores but with low validity (~20%). However, upon further training, the chemical validity of further checkpoints drops to 0% for the remaining runs, despite the overall DPO training and validation losses still dropping.
>
> With ERA, we see that we are able to similarly sample high QED small-molecules with reasonably high chemical validity (~85%). While the validity does drop over subsequent checkpoints, it does not do so precipitously. Moreover, the ERA-based alignment had no regularization (γ=0), and in the paper, we document how increasing γ can enable increases in chemical validity (main text Figure 4). Finally, we note that we did not extensively tune the hyperparameters for DPO , and it is possible that a different set of hyperparameters would elicit a more desired outcome; however, the lack of meaningful regularization in DPO [1], and its performance degradation in online metrics has been well-documented [2].
>
> [1] A General Theoretical Paradigm to Understand Learning from Human Preferences, Azar et al.
>
> [2] Preference Learning Algorithms Do Not Learn Preference Rankings, Chen et al.
>
>
>
>
>
> ## Binning
> This is an interesting suggestion. Given two samples (y, y') from our reference model, we desire that the relative weights under our policy converges to the true relative Boltzmann weight. If we rely on binning and assume that we have a single ranking of (y, y') in our dataset, the relative weights of y and y' under a model trained with DPO will not converge to the Boltzmann weight and will instead go to ∞. However, with ERA, we can ensure that the relative weights of y and y' will converge to the Boltzmann weight.
>
> As the reviewer states, one strength of ERA compared to DPO is that we can more easily avoid mode collapse (by tuning β), and also **independently** tune regularization towards a reference policy (γ), which is not possible with DPO. This ensures that we can promote desired sample diversity and similarity to a reference policy. We will reinforce this point in a revised version.

---

> > ### Comment · Reviewer_MXPc · 2024-08-12
> >
> > I thank the authors for their response and will maintain my score.

---

### Official Review · Reviewer_G1to · 2024-07-13

**Soundness:** 3
**Presentation:** 2
**Contribution:** 2
**Rating:** 5
**Confidence:** 3

**Summary:**

The authors study an important problem about searching through chemical space, where the number of possible molecules grows combinatorially with the number of atoms. They focus on aligning large autoregressive models trained on chemical compound databases to generate molecules. The energy rank alignment (ERA) algorithm is proposed to use an explicit reward function to produce a gradient-based objective for optimizing autoregressive policies.  The authors offer theoretical insights into the relationship between energy rank alignment (ERA) and proximal policy optimization (PPO), direct preference optimization (DPO). Their experiments show that ERA is scalable, does not require reinforcement learning, and performs well compared to DPO when preference observations per pairing are limited.

**Strengths:**

1. The authors study a significant problem about generating molecules with desired properties based on autoregressive models by proposing the energy rank alignment (ERA) algorithm.

2. This paper is well written.

3. The proposed methods work reasonably well.

**Weaknesses:**

1. Diversity, novelty and uniqueness are all important properties for drug discovery as discussed in previous works. To verify whether the models can be used to improve the process of drug discovery, the paper may benefit from comparing the aligned models with the reference model based on these metrics.

2. Missing the discussion of the related works which also focus on molecule optimization and drug discovery for both traditional and state-of-the-art methods, such as [1] [2] and so on.

3. The authors propose using reinforcement learning for drug optimization, a well-established method frequently employed in prior works, such as [3,4]. Additionally, advantage-based and multi-objective policy optimization are well-known in the reinforcement learning literature. A more comprehensive analysis of the limitations of this approach, along with a comparison to other existing methods, would have been beneficial.

[1] Drugassist: A large language model for molecule optimization.

[2] Automatic chemical design using a data-driven continuous representation of molecules.

[3] Optimization of molecules via deep reinforcement learning. Scientific Reports. 2019.

[4] Multi-constraint molecular generation based on conditional transformer, knowledge distillation and reinforcement learning. Nature Machine Intelligence. 2021.

**Questions:**

Please see above

**Limitations:**

Yes

---

> ### Author Rebuttal · Authors · 2024-08-05
>
> We thank the reviewer for their feedback and suggestions and for pointing us to additional works.
>
> ## Diversity Novelty and Uniqueness
> We investigate ERA on two tasks that mimic a drug-discovery effort and find that we are able to **efficiently** generate **novel, diverse, and unique compounds** that have high predicted biological activity according to ***in silico*** oracle functions (see global response). ERA consistently has the highest diversity compared with existing state-of-the-art methods.
>
> ## Discussion of and comparison to related works
> The approach described in [1] uses a Gaussian Process model to optimize molecular properties; however, this necessitates optimizing in a low-dimensional latent space, obtained with a VAE. With ERA and other RL methods described in the global response, we do not need a low-dimensional representation. We will add discussion of this method.
>
> We compare our approach to existing state-of-the-art methods (including some suggested by the reviewer) on the two tasks considered challenging and find that we are comparable to or better than existing methods, including on metrics related to diversity and sample efficiency (see global response).
>
> [1] Automatic Chemical Design Using a Data-Driven Continuous Representation of Molecules, Gomez-Bombarelli et al.

---

### Author Rebuttal · Authors · 2024-08-05

We appreciate the careful reviews and detailed feedback of our paper and address shared concerns and suggestions in this response.

## Simulated lead optimization and comparison to competitive approaches

The reviewers raised concerns about the complexity of the benchmarks included in the paper.
To address this concern, we now show the performance of ERA on two challenging tasks designed to mimic real-world lead optimization of small-molecules for specific biological targets. We find that our performance on these two tasks is **competitive with or better than existing state-of-the art methods** such as REINVENT [1] and MolRL-MGPT [2], which are both RL-based strategies.

We consider two targets, the kinases JNK3 and GSK3β, and aim to design small-molecules that are biologically active against each.
For each of these targets, we use an ***in silico*** oracle that predicts bioactivity, ranging from 0 to 1, where a higher value corresponds to stronger activity [3]. Using **only** data from ChemBL, we first carry out a short supervised fine-tuning step on all molecules in ChemBL with an oracle score above 0.5 (7386 molecules for JNK3 and 43381 for GSK3β). Using this fine-tuned model as our reference policy, we then align ($\beta$=100 and $\gamma$=0) as in Section 4.1, where we use a comparably high $\beta$ to target molecules with high activity.

From the aligned models, we sample 20k molecules (see rebuttal Figure 1a), and tabulate metrics of the top-100 novel molecules (see Table 1). We emphasize that the molecules in the top-100 are filtered to only include molecules that are distinct from any molecule in the ChemBL dataset and additionally that there are no repeated molecules in the top-100. For GSK3β, our mean score is marginally lower than the best performing method but the diversity in sampled molecules is significantly higher (i.e. lower IntDiv). For JNK3 our mean score is significantly higher than the best performing method **and** the diversity in sampled molecules is higher than any method. The inference costs are low for our approach; sampling 20k molecules and filtering steps takes only minutes.

We measure sample efficiency using the area under the curve (AUC) of top-K average property value versus the number of oracle calls top-K AUC [4]. We plot the top-10 average property value versus the number of oracle calls (see rebuttal Figure 1b) and report the top-10 AUC (see rebuttal Table 2). We only include novel molecules in this analysis; any sampled molecule that is in ChemBL, that has already been sampled, or that is invalid is discarded and additionally does **not** count towards an oracle call as these are filtered out before oracle evaluation.

We find that the top-10 AUC metric is higher than any competing method, demonstrating that our method can efficiently sample molecles with desired behavior. We note that once we reach a top-10 average of 1.0, we do not make futher oracle calls as subsequent oracle calls will not change the top-10 average and will artifiically inflate the AUC.

Our method is demonstrably better than existing state-of-the-art methods. With ERA, we generate both **novel** and **diverse** molecules with high predicted bioactivity, We also sample molecues with a high-oracle score compared to state-of-the-art methods **more efficiently**, ensuring that desired molecules can be generated with both a low inference cost and a low evaluation cost, the latter of which is important in settings where evaluation is expensive (e.g. wet-lab experiment).


[1] Reinvent 4: Modern AI-driven generative model design, Loeffler et al.

[2] De novo Drug Design using Reinforcement Learning with multiple GPT Agents, Hu et al.

[3] Excape-db: an integrated large scale dataset facilitating big data analysis in chemogenomics, Sun et al.

[4] Sample Efficiency Matters: A Benchmark for Practical Molecular Optimization, Gao et al.

## Comparison to Related Works

We will include more extensive discussion of related approaches suggested by the reviewers in our revision. We comment briefly here on methods that employ language models or transformers for molecular generation and use RL to optimize molecules and what distinguishes ERA. DrugAssist [1] uses human-machine dialogue with assistance from RL with human feedback (RLHF) and approaches multi-property optimization by incorporating diverse data streams. The ChemRLformer algorithm [2] leverages a text-based policy network and optimizes properties using a policy-gradient RL approach; however, in this framework, tuning sample diversity or regularization to a reference policy is challenging. The MGMG method [3] uses a knowledge-distilled conditional transformer and RL for multi-constraint optimization but involves many components and is cumbersome to optimize. The widely-used REINVENT method [4, 5] employs SMILES and multi-stage RL to tasks with mutliple reward models that vary in computational cost and accuracy. Finally, MolRL-MGPT [6] uses a multi-agent RL framework to promote sample diversity and has state-of-the-art performance, but training multiple GPT agents incurs significant training costs.


[1] DrugAssist: A Large Language Model for Molecule Optimization, Ye et al.

[2] Searching for High-Value Molecules Using Reinforcement Learning and Transformers, Ghugare et al.

[3] Multi-constraint molecular generation based on conditional transformer, knowledge distillation and reinforcement learning, Wang et al.

[4] REINVENT 2.0: An AI Tool for De Novo Drug Design, Blaschke et al.

[5] Reinvent 4: Modern AI-driven generative model design, Loeffler et al.

[6] De novo Drug Design using Reinforcement Learning with Multiple GPT Agents, Hu et al.

---

### Decision · Program_Chairs · 2024-09-25

**Decision:**

Reject

**Comment:**

In this work, authors propose a new algorithm, namely, energy rank alignment (ERA) for fine-tuning LLMs for molecule generation/optimization. All the reviewers have agreed that the work addresses an important and challenging problem. However, the reviewers have also raised concerns about limited analysis, more detailed discussions, evaluation metrics, inclusion of additional baselines and also more detailed discussions on the limitations and related works. Authors have addressed many of these concerns. However, several concerns regarding baselines and evaluations metrics still remain open. Hence, the manuscript in its present form cannot be recommended for acceptance.